# Revisiting Sharpness-Aware Minimization: A More Faithful and Effective Implementation

**Jianlong Chen, Zhiming Zhou**[*]
Key Laboratory of Interdisciplinary Research of Computation and Economics
Shanghai University of Finance and Economics
`chen.jianlong@stu.sufe.edu.cn, zhouzhiming@mail.sufe.edu.cn`

## Abstract

Sharpness-Aware Minimization (SAM) enhances generalization by minimizing the maximum training loss within a predefined neighborhood around the parameters. However, its practical implementation approximates this as gradient ascent(s) followed by applying the gradient at the ascent point to update the current parameters. This practice can be justified as approximately optimizing the objective by neglecting the (full) derivative of the ascent point with respect to the current parameters. Nevertheless, a direct and intuitive understanding of why using the gradient at the ascent point to update the current parameters works superiorly, despite being computed at a shifted location, is still lacking. Our work bridges this gap by proposing a novel and intuitive interpretation. We show that the gradient at the single-step ascent point, when applied to the current parameters, provides a better approximation of the direction from the current parameters toward the maximum within the local neighborhood than the local gradient. This improved approximation thereby enables a more direct escape from the maximum within the local neighborhood. Nevertheless, our analysis further reveals two issues. First, the approximation by the gradient at the single-step ascent point is often inaccurate. Second, the approximation quality may degrade as the number of ascent steps increases. To address these limitations, we propose in this paper eXplicit Sharpness-Aware Minimization (XSAM). It tackles the first by explicitly estimating the direction of the maximum during training, while addressing the second by crafting a search space that effectively leverages the gradient information at the multi-step ascent point. XSAM features a unified formulation that applies to both single-step and multi-step settings and only incurs negligible computational overhead. Extensive experiments demonstrate the consistent superiority of XSAM against existing counterparts across various models, datasets, and settings. Code is available at `https://github.com/Cccjl219/XSAM`.

## 1 Introduction

The success of modern machine learning relies heavily on overparameterization. This necessitates strong regularization, either implicit or explicit, from the training procedures (Srivastava et al., 2014; Gidel et al., 2019; Karakida et al., 2023) to ensure generalization beyond the training set (Zhang et al., 2021). In recent years, Sharpness-Aware Minimization (SAM) (Foret et al., 2020; Kwon et al., 2021; Liu et al., 2022b; Kim et al., 2023; Mordido et al., 2024) has attained significant attention for its potential to enhance the generalization of machine learning models, *in a direct optimization manner*.

SAM seeks to minimize the maximum training loss within a predefined neighborhood around the parameters, thereby promoting flatter minima and better generalization. Its effectiveness is evidenced by empirical successes across various domains (Bahri et al., 2021; Rangwani et al., 2022b;a; Fan et al., 2025). However, its practical implementation approximates this as: carry out one or a few steps of gradient ascent, and then apply the gradient from the ascent point to update the current parameters.

Though being justified as approximately optimizing the objective by neglecting the Jacobian matrix of the ascent point with respect to the current parameters (Foret et al., 2020), the underlying mechanism

---

[*]Corresponding author.

remains poorly understood. A body of research (Wen et al., 2023; Bartlett et al., 2023; Andriushchenko et al., 2023a; Andriushchenko & Flammarion, 2022; Andriushchenko et al., 2023b) has sought to demystify SAM after such approximations. However, a direct and intuitive understanding of why applying the *nonlocal* gradient at the ascent point to update the current parameter works superiorly is still lacking. This gap necessitates a deeper investigation into SAM's fundamental mechanisms, which motivates our work.

**Common misinterpretation**. A prevalent misunderstanding must be clarified before we proceed: applying the gradient at the estimated maximum point DOES NOT necessarily lead to the minimization of the maximum loss within the local neighborhood. *The key here is that there is a shift in location: the gradient is computed at the estimated maximum point, but applied to the current parameters.* The nuisance can be clear on considering the extreme case: the gradient at a point arbitrarily distant from the current parameters provides vanishingly little information about the local loss geometry.

To unravel the mystery of the SAM update, we commence by visualizing the local loss surface during SAM training. As shown in Figure 1a and further illustrated in Appendix A, our visualization analysis reveals an important underlying mechanism. Specifically, the gradient at the single-step ascent point, when applied to the current parameters, generally provides a better approximation of the direction from the current parameters toward the maximum within the local neighborhood than the gradient at the current parameters. Therefore, updating the current parameters along the direction opposite to the gradient at the single-step ascent point enables a more direct escape from the maximum. It thereby more effectively reduces the worst-case loss in the neighborhood, leading to improved generalization.

The above interpretation rationalizes the application of the gradient at the single-step ascent point to the current parameters. Nevertheless, our visualizations simultaneously reveal two limitations. First, the approximation by the gradient at the single-step ascent point is often inaccurate (as exemplified in Figure 1a). The approximation quality is also unstable, exhibiting large variations as the local loss landscape evolves (evidenced by further visualizations in Appendix A). Second, as illustrated by Figure 1b (and Figure 10 in Appendix A), the approximation quality may get worse as the number of ascent steps increases, explaining the unexpectedly inferior performance of multi-step SAM.

Motivated by these observations, we propose in this paper eXplicit Sharpness-Aware Minimization (XSAM), which fundamentally tackles the approximation inaccuracy issue of the SAM gradient by explicitly estimating the direction from the current parameters toward the maximum. This is achieved by probing the loss values in different directions at the neighborhood boundary. To ensure its high quality throughout training, XSAM dynamically updates this estimation.

Probing the entire high-dimensional neighborhood for estimating the direction can be computationally intractable. We therefore constrain the probe to a two-dimensional hyperplane spanned by the gradient at the final ascent point (i.e., the point reached after $k \geq 1$ ascent steps) and the vector from the current parameters to that point. This definition is crucial. It ensures that the point with the highest known loss, i.e., the one pointed to by the gradient at the final ascent point, lies within the hyperplane. Such a definition also simultaneously addresses the inaccuracy issue of directly applying the gradient at the multi-step ascent point to the current parameters, while fully leveraging its informational value.

We express the estimated direction in terms of the spherical interpolation factor of the two spanning vectors, which, according to our experiments, changes slowly during training. Therefore, it requires only infrequent updates and incurs negligible computational overhead. With this improved estimate of the direction toward the maximum, XSAM escapes the nearby high-loss regions more effectively, thereby achieving better generalization. Extensive experiments demonstrate that XSAM consistently outperforms existing counterparts across various models, datasets, and settings.

The primary contributions of this work are threefold:

- We provide a novel, intuitive interpretation of the fundamental mechanism of SAM: the gradient at the (single-step) ascent point offers a superior approximation of the direction from the current parameter toward the maximum within the local neighborhood than the local gradient; thereby, it enables a more direct escape from the maximum within the local neighborhood.

- Our analysis further reveals that the approximation by the gradient at the single-step ascent point is often inaccurate, and its quality varies largely during training. Moreover, the approximation quality may degrade as the number of ascent steps increases, explaining the inferior performance of multi-step SAM. These collectively demonstrate the sub-optimality of the SAM gradient.

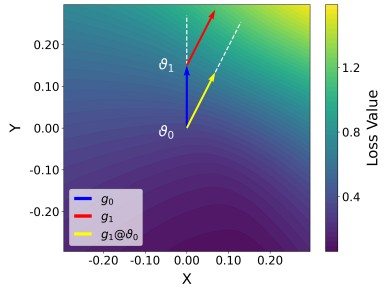

(a) Visualization of single-step SAM

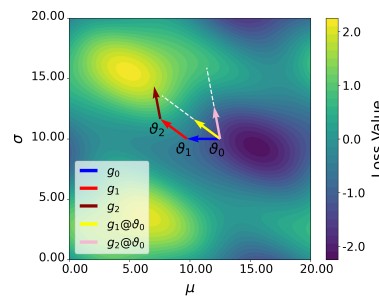

(b) Simulation of multi-step SAM

Figure 1: (a) Visualization of the local loss surface of single-step SAM[1] on the hyperplane spanned by the gradient $g_0$ at the current parameter $\vartheta_0$ and the gradient $g_1$ at the single-step ascent point $\vartheta_1$. $\vartheta_0$ is set as the origin, the $Y$-axis is defined along the direction of $g_0$, and the $X$-axis is aligned with the component of $g_1$ perpendicular to $g_0$. The visualized arrows of gradients are set to have length $\rho$. **We see that $g_1 @ \vartheta_0$ (i.e., $g_1$ applied to $\vartheta_0$) points clearly closer to the direction from $\vartheta_0$ toward the maximum within the local neighborhood than $g_0$.** The targeted direction is roughly from the origin to the upper-right corner in the figure. The loss along $g_1 @ \vartheta_0$ (i.e., $L(\vartheta_0 + \rho_m \cdot g_1 / \|g_1\|)$) is higher than that along $g_0$ (i.e., $L(\vartheta_0 + \rho_m \cdot g_0 / \|g_0\|)$), for sufficiently large $\rho_m$. (b) A simulation of multi-step SAM on a 2D test function. The approximation quality by the SAM gradient may get worse as the number of ascent steps increases. $g_2 @ \vartheta_0$ inferiorly identifies the direction from $\vartheta_0$ toward the maximum within the neighborhood (the upper-left high-loss region in yellow) than $g_1 @ \vartheta_0$.

- We propose XSAM, which addresses these limitations of SAM by explicitly estimating the direction from the current parameter toward the maximum, within a novel, principled search space during training. This leads to a more faithful and effective implementation of sharpness-aware minimization. Extensive experiments demonstrate the consistent superiority of XSAM.

## 2 REVISITING SHARPNESS-AWARE MINIMIZATION

This section reviews the objective of Sharpness-Aware Minimization (SAM) and its classical approximate optimization method, followed by our novel interpretation of its underlying mechanism.

### 2.1 THE OBJECTIVE AND CLASSICAL APPROXIMATION OF SAM

SAM (Foret et al., 2020) aims to find parameters that minimize the maximum training loss (i.e., worst-case loss) over a predefined $\rho$-neighborhood around the parameters. The formal objective is:

$$\min_\theta \max_{\|\delta\| \leq \rho} L(\theta + \delta), \tag{1}$$

where $L$ is the training loss, $\theta \in \mathbb{R}^n$ is the model parameters, and $\delta \in \mathbb{R}^n$ is the perturbation vector.[2]

Since exactly solving the inner maximization in Equation (1) is computationally expensive, SAM approximates it by performing one or a few steps of gradient ascent from the current parameters.

Assuming the procedure involves $k \geq 1$ successive gradient ascent steps, it proceeds as follows: initialize $\vartheta_0 = \theta$, and then for each step $i = 0, 1, \ldots, k-1$:

1) Compute the gradient at the current point $\vartheta_i$: $g_i = \nabla_{\vartheta_i} L(\vartheta_i)$;
2) Ascend along the direction of $g_i$ by a distance of $\rho_i$: $\vartheta_{i+1} = \vartheta_i + \rho_i \frac{g_i}{\|g_i\|}$.

This formulation unifies the single-step ($k = 1$) and multi-step ($k > 1$) settings, with the constraint $\sum_{i=0}^{k-1} \rho_i \leq \rho$ ensuring the total perturbation remains within the $\rho$-ball. The procedure yields the final perturbed parameters directly as $\vartheta_k$, while approximating the best perturbation $\delta^*$ as $\vartheta_k - \vartheta_0$.

---

[1] Data is collected at the first iteration of the 150th epoch in training ResNet-18 on CIFAR-100.
[2] For simplicity, we default all norms to $\ell_2$.

After such approximation of the best perturbation, the SAM objective in Equation (1) reduces to:

$$\min_\theta L(\theta + \delta^*), \qquad \text{or equivalently,} \qquad \min_\theta L(\vartheta_k). \qquad (2)$$

To optimize this objective efficiently, SAM employs a key approximation. It assumes $\nabla_\theta \delta^* = \mathbf{0}$, or equivalently, $\nabla_\theta \vartheta_k = I$, thereby avoiding involving expensive higher-order derivatives. Formally,

$$\nabla_\theta L(\theta + \delta^*) = \nabla_\theta L(\vartheta_k) = \nabla_{\vartheta_k} L(\vartheta_k) \cdot \underbrace{\nabla_\theta(\vartheta_k)}_{\text{Approximated as identity matrix } I} \approx \nabla_{\vartheta_k} L(\vartheta_k). \qquad (3)$$

The resulting algorithm essentially applies the gradient at the final ascent point $\vartheta_k$ to $\theta$:

$$\theta_{t+1} = \theta_t - \eta_t \cdot \nabla_{\vartheta_k} L(\vartheta_k). \qquad (4)$$

## 2.2 A Novel Interpretation of SAM's Underlying Mechanism

Despite the key approximation in the classical SAM algorithm being justified as assuming $\nabla_\theta \vartheta_k = I$, it leads to an unusual gradient operation, applying the gradient at another point $(\vartheta_k)$ to the current parameters $(\theta)$. It is apparent that applying the gradient at an arbitrarily distant point to the current parameters makes no sense, since it brings vanishingly little information about the local loss geometry around the current parameters. This contradiction raises a fundamental question: How is $\vartheta_k$ special? Why does applying this nonlocal gradient tend to outperform the local gradient in practice?

While a body of literature has sought to explain how SAM works after such approximation (Wen et al., 2023; Bartlett et al., 2023; Andriushchenko et al., 2023a;b), they often attribute it to implicit bias or regularization. None of them directly addresses our core inquiry: the underlying mechanism that enables this specific nonlocal gradient operation to be effective, which is the focus of this work.

### 2.2.1 Empirical Analysis through Visualizations

To unravel the underlying mechanism, we start by visualizing the gradients at the ascent point on the local loss surface during SAM training. For a tractable analysis and a clear comparison between the gradient at the ascent point and the gradient at the current parameters, we focus on the loss surface over the hyperplane spanned by these two gradient vectors. We begin with the single-step setting.

**Better Approximation**. As depicted in Figure 1a, the gradient at the single-step ascent point, when applied to the current parameters, can better approximate the direction toward the maximum within the local neighborhood than the gradient at the current parameters (i.e., the local gradient). More specifically, $g_1@\vartheta_0$ points clearly closer to the high-loss region around the upper-right corner than $g_0$, and the loss value along $g_1@\vartheta_0$ is also literally higher. This phenomenon is consistently observed in practice, as shown by additional visualizations in Appendix A.

**Inaccuracy and Instability**. Although $g_1@\vartheta_0$ provides a better approximation than $g_0$, we can clearly see in Figure 1a that the approximation by $g_1@\vartheta_0$ can still be rough and inaccurate. In fact, according to the additional visualizations in Appendix A, the approximation quality by $g_1@\vartheta_0$ is also unstable, exhibiting large variations during training. This suggests that such an approximation by $g_1@\vartheta_0$ can not well adapt to the evolving local loss landscape.

**Multi-Step Degradation**. We further extend the visualization analysis to multi-step settings. To approximate the complexity of high-dimensional landscapes, where multi-step ascent gradients deviate from a 2D plane, we simulate the process on a suitably complex 2D test function. As shown in Figure 1b, the gradient at the multi-step ascent point, when applied to the current parameters, may act as an unexpectedly poorer approximation compared to the gradient at the single-step ascent point. Specifically, $g_2@\vartheta_0$ inferiorly indicates the nearby high-loss region for $\vartheta_0$ than $g_1@\vartheta_0$. Notably, $g_2$ at its original position $\vartheta_2$ indeed points toward the nearby high-loss region; however, when applied to $\vartheta_0$, the resulting vector $g_2@\vartheta_0$ points toward a relatively flat region. This offers a visual explanation for why multi-step SAM does not work as well as expected (Foret et al., 2020; Andriushchenko & Flammarion, 2022). Additional simulation results supporting this finding are included in Appendix A.

### 2.2.2 Theoretical Confirmation under Second-Order Approximation

In this section, we substantiate our core empirical observations with the following results:

**Proposition 1.** *Let $L : \mathbb{R}^n \to \mathbb{R}$ be a twice continuously differentiable function that admits a second-order approximation at $\vartheta_0$ with:*

- *$\nabla L(\vartheta_0) = g_0$, which does not equal to 0;*
- *$\nabla L\left(\vartheta_0 + \rho \frac{g_0}{\|g_0\|}\right) = g_1$, which is not parallel to $g_0$;*
- *Hessian $H = \nabla^2 L(\vartheta_0)$ positive definite.*

*Then there exists $\rho_0 > 0$ such that for all $\rho_m > \rho_0$:*

1) SAM better approximates the direction toward the maximum in the vicinity than SGD

$$L\left(\vartheta_0 + \rho_m \frac{g_1}{\|g_1\|}\right) > L\left(\vartheta_0 + \rho_m \frac{g_0}{\|g_0\|}\right);$$

2) There exist better approximations than SAM *there exists $\alpha \in \mathbb{R}$ such that*

$$L\left(\vartheta_0 + \rho_m \frac{g_\alpha}{\|g_\alpha\|}\right) > L\left(\vartheta_0 + \rho_m \frac{g_1}{\|g_1\|}\right), \quad g_\alpha = \alpha g_1 + (1 - \alpha)g_0.$$

The first result in the proposition delivers that for any fixed distance that is relatively large, the loss along the direction of the gradient at the single-step ascent point is higher than that along the gradient at the current parameters. This confirms, from the loss-value perspective, that the gradient of single-step SAM better approximates the direction toward the maximum within its local neighborhood than that of SGD. Note that a relatively large distance is necessary for the second-order term to dominate the first-order term. For a distance that is too small, $g_0$ is by definition the steepest ascent direction. A detailed proof is provided in Appendix B. We additionally compare the losses of $L(\vartheta_0 + \rho_m\, g_1/\|g_1\|)$ and $L(\vartheta_0 + \rho_m\, g_0/\|g_0\|)$ across different $\rho$ and $\rho_m$ in actual experiments. See Figure 11 and 12 in Appendix A, which provides further empirical evidence of this result.

The second result in the proposition implies that there exist better approximations than the gradient of single-step SAM even in the two-dimensional hyperplane spanned by $g_0$ and $g_1$. This confirms our observation that the approximation by the gradient at the single-step ascent point is often inaccurate.

### 2.2.3 Heuristic Explanation and Deductive Analysis

To help establish a more intuitive understanding of why $g_1@\vartheta_0$ provides a better approximation for the direction of the maximum, we further provide the following heuristic explanation. Assuming the Hessian matrix of the loss function exhibits sufficiently slow variation within the local neighborhood, i.e., the gradient field evolves smoothly. Then, if $g_1$ is not parallel to $g_0$, the directional change from $g_0$ to $g_1$ reveals how the gradient field evolves in the surroundings. Considering additional *virtual* ascent steps within the local region, e.g., $\vartheta_2$ and $g_2$. The directional change from $g_1$ to $g_2$ will tend to follow a similar trend as that from $g_0$ to $g_1$. The same pattern persists for all subsequent virtual ascent steps, i.e., the virtual ascent trajectory will tend to curve in a consistent manner. Therefore, the high-loss region identified by the virtual ascent trajectory will likely be located at a position that is further shifted from the one-step ascent point $\vartheta_1$, along the direction of $g_1$, but curves further in the evolving direction of the gradient. Its direction relative to $\vartheta_0$ is thus better captured by $g_1@\vartheta_0$ than by $g_0$. Nevertheless, such an approximation is inherently inaccurate.

In multi-step settings, a crucial observation is that each adjacent pair of steps $(i, i+1)$ recapitulates the configuration of single-step SAM. Consequently, the conclusion from the single-step analysis holds inductively for each step. That is, $g_{i+1}@\vartheta_i$ better approximates the direction toward the maximum than $g_i@\vartheta_i$, for $i \in [0, \ldots, k-1]$. However, a critical discrepancy arises in multi-step SAM: it directly applies $g_k$ to $\vartheta_0$, but it remains unclear whether $g_k@\vartheta_0$ stands as a better approximation of the direction from $\vartheta_0$ toward the maximum than $g_1@\vartheta_0$ (or even $g_0$). The core difference here is that $g_1$ is evaluated along the ray defined by $g_0$ and $\vartheta_0$, whereas $g_k$ may substantially deviate from the ray defined by $g_0$ and $\vartheta_0$. Because the entire multi-step trajectory can curve significantly. This renders the direct application of $g_k$ to $\vartheta_0$ potentially suboptimal or unjustified.

As a final remark, a simple deduction reveals the inherent inaccuracy of the SAM gradient approximation: Consider SAM operating on a fixed loss surface. Regardless of how accurately $g_k@\vartheta_0$ currently

**Algorithm 1** XSAM

**Input:** Initial parameters $\theta_0$, number of iterations $T$, number of ascent steps $k \geq 1$, perturbation radius $\{\rho_i\}$, neighborhood radius $\rho_m$, $\alpha^*$ update frequency $T_\alpha$, learning rate $\{\eta_t\}$
**Output:** Final parameters $\theta_T$
1: **for** $t = 0$ **to** $T - 1$ **do**
2:     $\vartheta_0 = \theta_t$
3:     **for** $i = 0$ **to** $k - 1$ **do**  ▷ *Single-step: $k = 1$*
4:         $g_i = \nabla_{\vartheta_i} L(\vartheta_i)$
5:         $\vartheta_{i+1} = \vartheta_i + \rho_i \frac{g_i}{\|g_i\|}$
6:     **end for**
7:     $g_k = \nabla_{\vartheta_k} L(\vartheta_k)$
8:     $v_0 = \frac{\vartheta_k - \vartheta_0}{\|\vartheta_k - \vartheta_0\|}, \quad v_1 = \frac{g_k}{\|g_k\|}$
9:     $\psi = \arccos(v_0 \cdot v_1)$
10:     **if** $t \bmod T_\alpha = 0$ **then**
11:         $\alpha_t^* = \arg\max_\alpha L(\vartheta_0 + \rho_m \cdot v(\alpha))$,
12:         where $v(\alpha) = \frac{\sin((1-\alpha)\psi)}{\sin(\psi)}v_0 + \frac{\sin(\alpha\psi)}{\sin(\psi)}v_1$
13:     **else**
14:         $\alpha_t^* = \alpha_{t-1}^*$
15:     **end if**
16:     $\theta_{t+1} = \theta_t - \eta_t \cdot v(\alpha_t^*) \cdot \|g_k\|$
17: **end for**

Table 1: Training time comparison. Values are presented as hours/200 epochs, SAM / XSAM.

|  | CIFAR-10 | CIFAR-100 | Tiny-ImageNet |
|---|---|---|---|
| VGG-11 | 0.93 / 0.96 | 0.98 / 1.03 | 2.18 / 2.22 |
| ResNet-18 | 2.35 / 2.39 | 2.40 / 2.43 | 4.95 / 4.98 |
| DenseNet-121 | 8.02 / 8.08 | 8.05 / 8.07 | 16.50 / 16.55 |

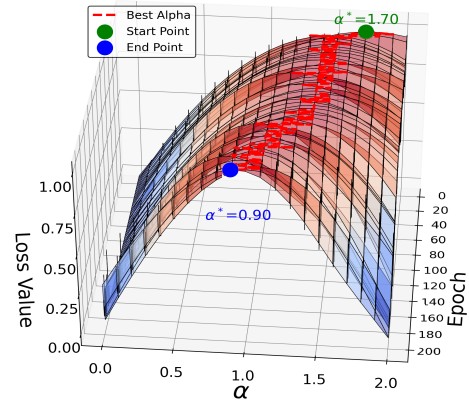

Figure 2: Slow variation of $\alpha^*$ during training.

approximates the direction, as long as we continuously decrease $\{\rho_i\}$ (for all $i \in [0, k-1]$) toward 0, $g_k$ will reduce to $g_0$. Consequently, the approximation quality of $g_k@\vartheta_0$ will get reduced arbitrarily close to that of the original gradient $g_0$. This sensitivity to the choice of $\{\rho_i\}$ also implies that, for an arbitrary $\{\rho_i\}$, it is typically suboptimal (even for a certain fixed loss surface).  ▷ On the other hand, we can also tune $\{\rho_i\}$ to make it the best possible approximation, which could have played a role in the practical effectiveness of SAM. Nevertheless, given the evolving local loss landscape during training, the approximation with any fixed $\{\rho_i\}$ can hardly remain relatively accurate throughout.

## 3 EXPLICIT SHARPNESS-AWARE MINIMIZATION

As shown in the above section, the approximation by the SAM gradient is often inaccurate and lacks adaptivity to the evolving local loss landscape. Moreover, the approximation quality may degrade as the number of ascent steps increases. To provide an integrated solution that simultaneously addresses all these limitations, we propose in this section eXplicit Sharpness-Aware Minimization (XSAM).

XSAM addresses the inaccuracy issue by explicitly probing the location of the maximum within the local neighborhood, thereby providing a more accurate update direction. By dynamically performing this probe during training, it further enhances adaptivity to the evolving local loss landscape.

Probing the maximum within the entire high-dimensional neighborhood can be computationally intractable. We therefore assume that the maximum is located at the neighborhood boundary, while further constraining the probe to a two-dimensional hyperplane. The 2D hyperplane is spanned by the gradient at the final ascent point (i.e., the point reached after $k \geq 1$ ascent steps) and the vector from the current parameters to that point. Formally, the two spanning vectors are defined as:

$$v_0 = \frac{\vartheta_k - \vartheta_0}{\|\vartheta_k - \vartheta_0\|}, \qquad v_1 = \frac{g_k}{\|g_k\|}. \tag{5}$$

This definition of the two-dimensional hyperplane is crucial and provides four key advantages. First, it ensures that the point with the highest known loss (the one pointed to by $g_k$, standing at $\vartheta_k$) lies within the hyperplane. Second, it avoids the inaccuracy issue of directly applying the gradient at the multi-step ascent point to the current parameters, while fully leveraging its informational value. Specifically, we use $\vartheta_k$ and $g_k$ to define a search space that encompasses all the information they contain, instead of directly applying $g_k$ to $\vartheta_0$. Third, it offers a unified formulation for both single-step and multi-step settings. Note that when $k = 1$, $v_0$ and $v_1$ correspond to the directions

of $g_0$ and $g_1$, respectively. Fourth, normalization is applied to separate direction from magnitude, allowing us to manage them independently.

To probe within the two-dimensional hyperplane, we generate new directions as the spherical linear interpolation between $v_0$ and $v_1$:

$$v(\alpha) = \frac{\sin((1-\alpha)\psi)}{\sin(\psi)}v_0 + \frac{\sin(\alpha\psi)}{\sin(\psi)}v_1, \tag{6}$$

where $\psi = \arccos(v_0 \cdot v_1)$ and $\alpha$ is the interpolation factor. It has $\|v(\alpha)\| = 1$ for any $\alpha$, $v(0) = v_0$, $v(1) = v_1$. More generally, $v(\alpha)$ is a unit vector that rotates from $v_0$ by an angle of $\alpha \cdot \psi$ along the direction toward $v_1$. It can span all possible directions in the search space.

We then determine the direction, parametrized by $\alpha^*$, that maximizes the loss at a predefined distance:

$$\alpha^* = \arg \max_{\alpha \in [0,a]} L(\vartheta_0 + \rho_m \cdot v(\alpha)), \tag{7}$$

where $\rho_m$ is a hyperparameter specifying the radius of the *true* (in contrast to the perturbation radius) sharpness-aware neighborhood. In each dynamic search, we uniformly sample $\alpha$ values from $[0, a]$. In practice, setting $a$ to 2 or 4 and sampling 20–40 samples is typically sufficient.

Once $\alpha^*$ is identified, the model parameters are updated using $-v(\alpha^*)$ as the descent direction. The gradient scale, by default, is set to $\|g_k\|$ to make it consistent with SAM[3]. Formally,

$$\theta_{t+1} = \theta_t - \eta_t \cdot v(\alpha^*) \cdot \|g_k\|, \tag{8}$$

by which $v(\alpha^*)$ steers the parameters away from the estimated maximum within the neighborhood.

**Faithfulness and Effectiveness.** Since we use $L(\vartheta_0 + \rho_m \cdot v(\alpha))$ as a proxy[4], the method explicitly identifies the maximum within a neighborhood of radius $\rho_m$. Although restricted to a hyperplane, this approximation relies only on the boundary assumption. It thus more faithfully identifies the maximum in the local neighborhood, in contrast to directly regarding $\vartheta_k$ as the maximum or approximating its direction by $g_k @ \vartheta_0$. XSAM thereby more authentically realizes the sharpness-aware minimization.

**The Cost of Explicit Estimation**. The evaluation of each $\alpha$ requires a forward pass. Thus, the cost of explicit estimation scales with the number of sampled $\alpha$ values times the cost of a forward pass. If performed at every iteration, this would introduce substantial overhead. Fortunately, frequent updates of $\alpha^*$ are unnecessary. Our experiments show that $\alpha^*$ remains relatively stable and varies smoothly during training (Figure 2). By default, we adopt an epoch-wise update strategy: $\alpha^*$ is updated at the first iteration of each epoch and then fixed for the remainder. Runtime comparison is shown in Table 1, indicating the additional overhead is negligible. Further details are provided in Appendix C.

## 4 RELATED WORK

SAM has been extended in several distinct directions. One line of work focuses on improving the gradient ascent (i.e., perturbation) step, addressing issues such as parameter scale dependence (ASAM (Kwon et al., 2021); Fisher SAM (Kim et al., 2022)), approximation quality (RSAM (Liu et al., 2022b); CR-SAM (Wu et al., 2024)), and perturbation stability (VaSSO (Li & Giannakis, 2024); FSAM (Li et al., 2024)). These approaches are largely complementary to ours; for instance, Appendix E.1 demonstrates that integrating XSAM with ASAM yields additional performance gains.

Another line of research targets the parameter update step. GSAM (Zhuang et al., 2022) combines the perturbed gradient with the orthogonal component of the local gradient. GAM (Zhang et al., 2023) simultaneously optimizes empirical loss and first-order flatness. In particular, WSAM (Yue et al., 2023) and Zhao et al. (2022a) derive their update rules as a linear combination of $g_0$ and $g_1$ through weighted sharpness regularization and gradient-norm penalization, respectively. While their superior performance over SAM is readily explained by our interpretation, this very perspective reveals a critical weakness: their dependence on a fixed combination weight, treated as a hyperparameter, is inherently suboptimal. In contrast, XSAM explicitly estimates the optimal interpolation factor

---

[3]Alternative gradient scaling strategies are examined in Appendix F.

[4]Our implementation uses only the current batch, consistent with the standard SAM procedure.

Table 2: Test accuracies on classification tasks in the single-step setting.

| Dataset | CIFAR-10 | | | CIFAR-100 | | | Tiny-ImageNet | | |
|---|---|---|---|---|---|---|---|---|---|
| Model | VGG-11 | ResNet-18 | DenseNet-121 | VGG-11 | ResNet-18 | DenseNet-121 | VGG-11 | ResNet-18 | DenseNet-121 |
| SGD | $93.19_{\pm0.11}$ | $96.15_{\pm0.05}$ | $96.34_{\pm0.11}$ | $71.46_{\pm0.17}$ | $78.55_{\pm0.20}$ | $81.78_{\pm0.06}$ | $47.44_{\pm0.33}$ | $57.02_{\pm0.42}$ | $61.93_{\pm0.10}$ |
| SAM | $93.83_{\pm0.06}$ | $96.59_{\pm0.06}$ | $96.97_{\pm0.02}$ | $74.01_{\pm0.05}$ | $80.93_{\pm0.11}$ | $83.81_{\pm0.02}$ | $51.96_{\pm0.26}$ | $62.81_{\pm0.09}$ | $66.31_{\pm0.09}$ |
| XSAM | $\mathbf{94.25}_{\pm0.14}$ | $\mathbf{96.74}_{\pm0.04}$ | $\mathbf{97.15}_{\pm0.03}$ | $\mathbf{74.21}_{\pm0.14}$ | $\mathbf{81.24}_{\pm0.07}$ | $\mathbf{83.96}_{\pm0.10}$ | $\mathbf{52.58}_{\pm0.38}$ | $\mathbf{63.82}_{\pm0.23}$ | $\mathbf{66.81}_{\pm0.08}$ |

(a)       (b)       (c)

Figure 3: (a) Training trajectory comparisons on 2D test function. (b)-(c) Test accuracy comparisons of ResNet-18 trained on CIFAR-100 in single-step and multi-step ($k = 3$) settings with varying $\rho$.

dynamically during training and naturally extends this principle to multi-step settings. More fundamentally, our approach is derived from a reformulation of the sharpness-aware objective itself, rather than introducing an auxiliary regularization term, offering a more general and principled solution.

Multi-step SAM variants are discussed in Section 5.3, while additional related work on topics such as flatness, efficiency, and long-tail learning is deferred to Appendix G.

## 5 EMPIRICAL RESULTS

In this section, we empirically compare SAM and its related variants with the proposed XSAM. Due to space limitations, detailed experimental settings are deferred to Appendix D.

### 5.1 2D TEST FUNCTION

Following (Yue et al., 2023; Kim et al., 2022), we first evaluate methods on a 2D function featuring a sharp and a flat minimum within a certain distance, serving as an ideal testbed for sharpness-aware minimization. We compare SGD, SAM, and XSAM across different initial points and hyperparameters. XSAM consistently converges to the flat minima when $\rho_m$ is sufficiently large, whereas SAM and SGD are more prone to get trapped in the sharp minima. Representative training trajectories for each method are shown in Figure 3a. Both SAM and XSAM are evaluated in their single-step form.

### 5.2 EVALUATION UNDER THE SINGLE-STEP SETTING

In this section, we evaluate the methods under the single-step setting across a variety of classification datasets and model architectures. To stress-test the methods, we first tune SAM's learning rate, weight decay, and $\rho$ to achieve its optimal performance on each dataset. Other methods are then tuned using the same hyperparameters whenever feasible. To isolate the effect of different gradient directions and eliminate the influence of gradient scaling, all methods adopt SAM's gradient scale, i.e., $\|g_k\|$.

We evaluate the methods across diverse neural network architectures and datasets to ensure broad applicability. The experiments cover architectures ranging from VGG-11 (Simonyan & Zisserman, 2014) and ResNet-18 (He et al., 2016) to DenseNet-121 (Huang et al., 2017), encompassing classic models of increasing capacity. The datasets include CIFAR-10, CIFAR-100, and Tiny-ImageNet, which span increasing complexities. As shown in Table 2, SAM consistently outperforms SGD, confirming the superiority of the gradient direction of $g_1$ compared to $g_0$. Meanwhile, XSAM consistently outperforms SAM, highlighting the benefit of explicitly estimating the direction.

To provide a more thorough comparison, we evaluate performance under varying $\rho$ on CIFAR-100 using ResNet-18. For this experiment, we further include a WSAM-like baseline, which implements our method with a fixed but tunable $\alpha$, to highlight the benefit of dynamically estimating $\alpha$ compared to a static choice. The best fixed $\alpha$ for the WSAM is determined via grid search over $[-1.0, 3.0]$ with a step size of $0.25$. As shown in Figure 3b, the WSAM improves over SAM, while XSAM consistently achieves further and significant improvements over the WSAM.

Having established XSAM's potent performance under varying $\rho$, we further assess XSAM's generality on larger-scale and more diverse tasks. We conduct experiments on ImageNet with ResNet-50, a neural machine translation task with a Transformer (Vaswani et al., 2017), and CIFAR-100 with ViT-Ti (Dosovitskiy et al., 2020). The results in Table 3 show that XSAM consistently outperforms SAM, demonstrating its broad applicability across diverse tasks and models.

Table 3: Comparison of SAM and XSAM on larger-scale and more diverse tasks.

|  | ImageNet ResNet-50 (Accuracy) | Transformer IWSLT2014 (BLEU) | ViT-Ti CIFAR-100 (Accuracy) |
|---|---|---|---|
| SAM | $77.04 \pm 0.09$ | $35.30 \pm 0.04$ | $67.80 \pm 0.22$ |
| XSAM | $\mathbf{77.22 \pm 0.07}$ | $\mathbf{35.63 \pm 0.12}$ | $\mathbf{68.32 \pm 0.18}$ |

Table 4: Multi-step results on CIFAR-100 with ResNet-18. $\rho = \rho^*/k$ with $\rho^*$ for single-step.

| Methods | $k = 1$ | $k = 2$ | $k = 4$ |
|---|---|---|---|
| SAM | $80.93 \pm 0.11$ | $80.91 \pm 0.10$ | $80.65 \pm 0.26$ |
| LSAM | $80.93 \pm 0.11$ | $80.94 \pm 0.09$ | $80.74 \pm 0.18$ |
| LSAM+ | $80.61 \pm 0.20$ | $80.83 \pm 0.11$ | $80.41 \pm 0.03$ |
| MSAM | $80.93 \pm 0.11$ | $81.18 \pm 0.06$ | $81.01 \pm 0.09$ |
| MSAM+ | $80.83 \pm 0.05$ | $80.86 \pm 0.34$ | $80.77 \pm 0.08$ |
| XSAM | $\mathbf{81.27 \pm 0.07}$ | $\mathbf{81.44 \pm 0.09}$ | $\mathbf{81.37 \pm 0.24}$ |

## 5.3 Evaluation under the Multi-Step Setting

We proceed to evaluate and compare methods in a multi-step setting. We use a constant perturbation magnitude $\rho$ for all steps (i.e., $\rho_i = \rho$ for all $i$), therefore omitting the subscript $i$ for clarity. All experiments in this section are conducted on CIFAR-100 using a ResNet-18.

As the first experiment, we compare XSAM with multi-step SAM variants across different values of $k$. The considered methods include: MSAM (Kim et al., 2023), which updates parameters with $\sum_{i=1}^{k} g_i$, and LSAM (Mordido et al., 2024), which employs $\sum_{i=1}^{k} g_i/\|g_i\|$. To ensure a thorough comparison, we further introduce two augmented variants that incorporate the initial gradient $g_0$: MSAM+ ($\sum_{i=0}^{k} g_i$) and LSAM+ ($\sum_{i=0}^{k} g_i/\|g_i\|$). Consistent with our previous protocol, we isolate the effect of gradient direction by readjusting all gradients to have the norm $\|g_k\|$. The perturbation radius is set to $\rho = \rho^*/k$, where $\rho^*$ is the optimal value for single-step SAM, as suggested by Kim et al. (2023); all other hyperparameters remain unchanged from the single-step setup.

As shown in Table 4, the performance of SAM tends to decline as $k$ increases. This phenomenon can be attributed to the growing deviation of $g_k$ from the original ascent direction $g_0 @ \vartheta_0$ as the single ascent step is subdivided. As a result, when applied to $\vartheta_0$, it leads to a poorer approximation of the direction toward the maximum in the vicinity. In contrast, XSAM is not affected by this issue and typically benefits from more steps, demonstrating its superior ability to leverage multi-step ascent.

LSAM and MSAM, which incorporate intermediate ascent gradients ($g_i$ for $0 \leq i \leq k$), generally surpass SAM. The decline in SAM's performance with large $k$ suggests substantial deviation of $g_k$ from the ideal direction, which makes earlier, less-deviated gradients $g_i$ valuable. Notably, LSAM+, which essentially moves away directly from the identified maximum point by multi-step ascent, even underperforms SAM. This highlights the value of an extra explicit estimation of the direction toward the maximum. Nevertheless, XSAM consistently outperforms all these methods across all settings.

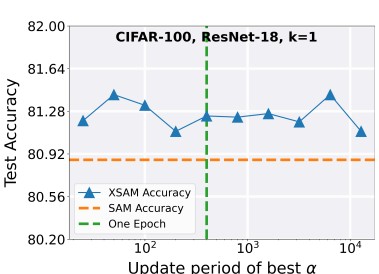

Figure 4: XSAM robustness to the $\alpha^*$ update frequency.

We further evaluate SAM and XSAM under a multi-step setting ($k = 3$) with a varying perturbation radius $\rho$. A multi-step extension of WSAM, which combines the gradients $g_k$ and $g_0$ with a fixed interpolation factor, is also compared. The results in Figure 3c indicate that while the WSAM variant outperforms SAM, XSAM consistently outperforms WSAM.

Figure 4 shows the robustness of XSAM to the $\alpha^*$ update frequencies. We observe no consistent pattern in performance when varying the update frequency of $\alpha^*$. Additional ablation results are presented in Appendix E.4. Appendix E.5 further visualizes the loss surface at convergence, illustrating that XSAM finds **flatter minima** than SAM.

## 6 CONCLUSION

In this paper, we have studied the underlying mechanism of SAM and provided a novel, intuitive explanation of why it is valid and effective to apply the gradient at the ascent point to the current parameters. We have shown that the SAM gradient in its single-step version can provably better approximate the direction of the maximum within the local neighborhood than that of SGD. We have further demonstrated that such an approximation can be inaccurate, lacks adaptivity to the evolving local loss landscape, and may degrade as the number of ascent steps increases. To address these limitations, we have proposed XSAM that explicitly and dynamically estimates the direction of the maximum within the local neighborhood during training. XSAM thereby more faithfully and effectively moves the current parameters away from high-loss regions. Extensive experiments across various models, datasets, tasks, and settings have demonstrated the effectiveness of XSAM.

## ACKNOWLEDGMENTS

This work was supported in part by the National Natural Science Foundation of China under Grant No. U22B2020.

## REPRODUCIBILITY STATEMENT

We have provided the code as supplementary material, along with detailed instructions for reproducing our experiments. The experimental settings and hyperparameters are described in the Appendix D. The datasets used in this paper are publicly available and can be downloaded online. Detailed proofs of the proposed proposition are included in the Appendix B.

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

APPENDIX

# A  VISUALIZATION OF LOSS SURFACE DURING TRAINING

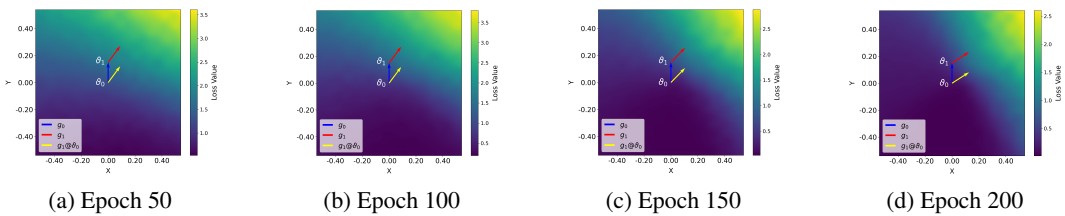

(a) Epoch 50          (b) Epoch 100          (c) Epoch 150          (d) Epoch 200

Figure 5: Visualization of loss surface during training: VGG-11 trained on CIFAR-100.

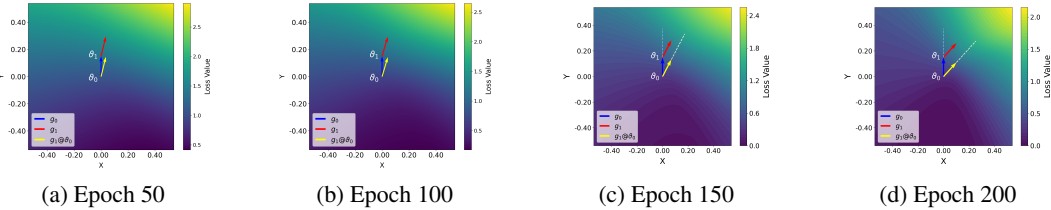

(a) Epoch 50          (b) Epoch 100          (c) Epoch 150          (d) Epoch 200

Figure 6: Visualization of loss surface during training: ResNet-18 trained on CIFAR-100.

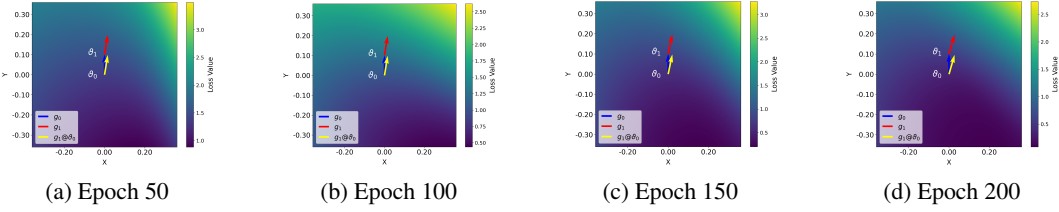

(a) Epoch 50          (b) Epoch 100          (c) Epoch 150          (d) Epoch 200

Figure 7: Visualization of loss surface during training: ViT-Ti trained on CIFAR-100.

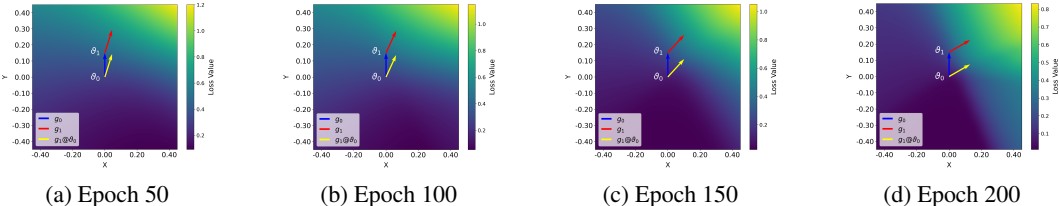

(a) Epoch 50          (b) Epoch 100          (c) Epoch 150          (d) Epoch 200

Figure 8: Visualization of loss surface during training: ResNet-18 trained on CIFAR-10.

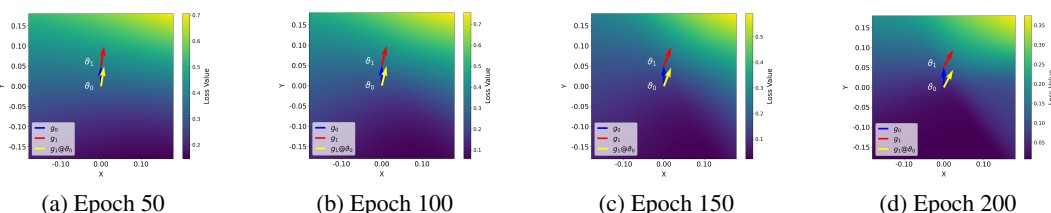

(a) Epoch 50          (b) Epoch 100          (c) Epoch 150          (d) Epoch 200

Figure 9: Visualization of loss surface during training: DenseNet-121 trained on CIFAR-10.

In this section, we provide more visualizations of the loss surfaces of different datasets and models during SAM training. The results are shown in Figure 5, 6, 7, 8, 9, and 10. The gradient of the ascent

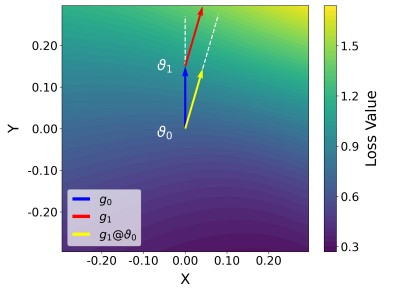

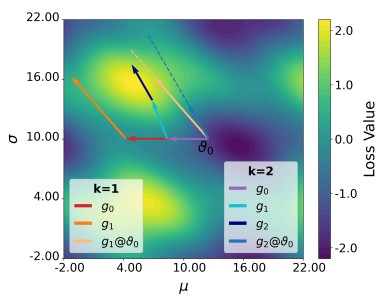

(a) Visualization of single-step SAM

(b) Simulation of multi-step SAM

Figure 10: (a) Visualization of the local loss surface of single-step SAM. The visualization procedure follows the same steps as in Figure 1a. Data is collected at the first iteration of the 100th epoch in training ResNet-18 on CIFAR-100. **We see that $g_1@\vartheta_0$ (i.e., $g_1$ applied to $\vartheta_0$) points clearly closer to the direction from $\vartheta_0$ toward the maximum within the local neighborhood than $g_0$.** he targeted direction is roughly from the origin to the upper-right corner in the figure. The loss along $g_1@\vartheta_0$ (i.e., $L(\vartheta_0 + \rho_m \cdot g_1/\|g_1\|)$) is higher than that along $g_0$ (i.e., $L(\vartheta_0 + \rho_m \cdot g_0/\|g_0\|)$), for sufficiently large $\rho_m$. (b) A simulation of multi-step SAM on a test function. The gradient at the multi-step ascent point, when applied to the current parameters, may be an inferior approximation of the direction toward the maximum.

point better approximates the direction toward the maximum within the neighborhood than the local gradient. However, the approximation can often be inaccurate and unstable during training.

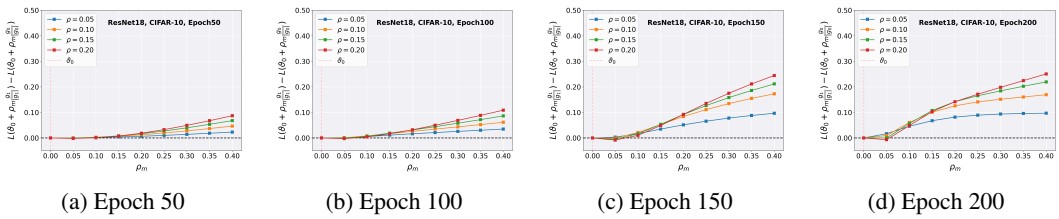

(a) Epoch 50      (b) Epoch 100      (c) Epoch 150      (d) Epoch 200

Figure 11: Visualization of $L(\vartheta_0 + \rho_m g_1/\|g_1\|) - L(\vartheta_0 + \rho_m g_0/\|g_0\|)$ during training.

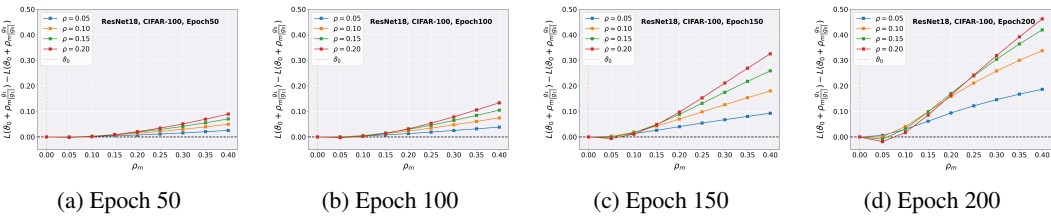

(a) Epoch 50      (b) Epoch 100      (c) Epoch 150      (d) Epoch 200

Figure 12: Visualization of $L(\vartheta_0 + \rho_m g_1/\|g_1\|) - L(\vartheta_0 + \rho_m g_0/\|g_0\|)$ during training.

We compare $L(\vartheta_0 + \rho_m g_1/\|g_1\|)$ and $L(\vartheta_0 + \rho_m g_0/\|g_0\|)$ across different $\rho$ and $\rho_m$ in Figure 11 and 12. We gradually increase $\rho_m$ for each $\rho$, while keeping $\vartheta_0$ and $g_0$ fixed. As can be seen, $L(\vartheta_0 + \rho_m g_1/\|g_1\|)$ becomes larger than $L(\vartheta_0 + \rho_m g_0/\|g_0\|)$ when $\rho_m$ is relatively large. This provides further evidence for our claim that along $g_1$ one can find a higher loss than long $g_0$ for $\vartheta_0$.

## B  PROOFS

**Proposition 1.** *Let $L : \mathbb{R}^n \to \mathbb{R}$ be a twice continuously differentiable function that admits a second-order approximation at $\vartheta_0$ with:*

- $\nabla L(\vartheta_0) = g_0$, *which does not equal to 0;*
- $\nabla L\left(\vartheta_0 + \rho \frac{g_0}{\|g_0\|}\right) = g_1$, *which is not parallel to $g_0$;*
- *Hessian $H = \nabla^2 L(\vartheta_0)$ positive definite.*

*Then there exists $\rho_0 > 0$ such that for all $\rho_m > \rho_0$:*

1) | *SAM better approximates the direction toward the maximum in the vicinity than SGD* |

$$L\left(\vartheta_0 + \rho_m \frac{g_1}{\|g_1\|}\right) > L\left(\vartheta_0 + \rho_m \frac{g_0}{\|g_0\|}\right);$$

2) | *There exist better approximations than SAM* | *there exists $\alpha \in \mathbb{R}$ such that*

$$L\left(\vartheta_0 + \rho_m \frac{g_\alpha}{\|g_\alpha\|}\right) > L\left(\vartheta_0 + \rho_m \frac{g_1}{\|g_1\|}\right), \quad g_\alpha = \alpha g_1 + (1-\alpha)g_0.$$

## B.1 PROOF OF THE FIRST CONCLUSION

*Proof.*

1. Since $L$ admits a second-order approximation at $\theta_0$:

$$L\left(\vartheta_0 + \rho_m \frac{g_1}{\|g_1\|}\right) = L(\vartheta_0) + \rho_m \frac{g_0^\top g_1}{\|g_1\|} + \frac{\rho_m^2}{2} \frac{g_1^\top H g_1}{\|g_1\|^2} + o(\rho_m^2),$$

$$L\left(\vartheta_0 + \rho_m \frac{g_0}{\|g_0\|}\right) = L(\vartheta_0) + \rho_m \|g_0\| + \frac{\rho_m^2}{2} \frac{g_0^\top H g_0}{\|g_0\|^2} + o(\rho_m^2).$$

2. For sufficiently large $\rho_m$, the $\rho_m^2$ term dominates. Thus, we need to show:

$$\frac{g_1^\top H g_1}{\|g_1\|^2} > \frac{g_0^\top H g_0}{\|g_0\|^2}.$$

3. Expand $g_1$ as the gradient of $L$ (which admits a second-order approximation) at $\vartheta_0 + \rho \frac{g_0}{\|g_0\|}$:

$$g_1 = g_0 + \rho H \frac{g_0}{\|g_0\|} + o(\rho).$$

4. Compute the numerator and denominator to the second order:

$$g_1^\top H g_1 = \left(g_0 + \rho H \frac{g_0}{\|g_0\|} + o(\rho)\right)^\top H \left(g_0 + \rho H \frac{g_0}{\|g_0\|} + o(\rho)\right)$$

$$= g_0^\top H g_0 + 2\rho \frac{g_0^\top H^2 g_0}{\|g_0\|} + \rho^2 \frac{g_0^\top H^3 g_0}{\|g_0\|^2} + o(\rho^2),$$

$$\|g_1\|^2 = \left\|g_0 + \rho H \frac{g_0}{\|g_0\|} + o(\rho)\right\|^2 = \|g_0\|^2 + 2\rho \frac{g_0^\top H g_0}{\|g_0\|} + \rho^2 \frac{g_0^\top H^2 g_0}{\|g_0\|^2} + o(\rho^2).$$

4. Ignoring higher-order terms $o(\rho^2)$, the inequality becomes:

$$\frac{g_0^\top H g_0 + 2\rho \frac{g_0^\top H^2 g_0}{\|g_0\|} + \rho^2 \frac{g_0^\top H^3 g_0}{\|g_0\|^2}}{\|g_0\|^2 + 2\rho \frac{g_0^\top H g_0}{\|g_0\|} + \rho^2 \frac{g_0^\top H^2 g_0}{\|g_0\|^2}} > \frac{g_0^\top H g_0}{\|g_0\|^2}.$$

5. Multiply both sides by the positive denominators (since $H$ is positive definite):

$$\left(g_0^\top H g_0 + 2\rho \frac{g_0^\top H^2 g_0}{\|g_0\|} + \rho^2 \frac{g_0^\top H^3 g_0}{\|g_0\|^2}\right) \|g_0\|^2 > g_0^\top H g_0 \left(\|g_0\|^2 + 2\rho \frac{g_0^\top H g_0}{\|g_0\|} + \rho^2 \frac{g_0^\top H^2 g_0}{\|g_0\|^2}\right).$$

6. Cancel common terms and divide by $\rho > 0$:

$$2\left(\|g_0\| g_0^\top H^2 g_0 - \frac{(g_0^\top H g_0)^2}{\|g_0\|}\right) + \rho\left(g_0^\top H^3 g_0 - \frac{g_0^\top H g_0 g_0^\top H^2 g_0}{\|g_0\|^2}\right) > 0.$$

7. Term verification:

- First term:
$$\|g_0\|^2 g_0^\top H^2 g_0 - (g_0^\top H g_0)^2 > 0.$$

This follows from the strict Cauchy-Schwarz inequality for the inner product, since $g_0$ and $Hg_0$ are not parallel by assumption.

- Second term:
$$\|g_0\|^2 g_0^\top H^3 g_0 - g_0^\top H g_0 g_0^\top H^2 g_0 \geq 0.$$

Let $H = \sum_{i=1}^n \lambda_i v_i v_i^\top$ be the spectral decomposition with $\lambda_i > 0$. Expressing $g_0 = \sum_{i=1}^n \alpha_i v_i$:

$$\|g_0\|^2 g_0^\top H^3 g_0 - g_0^\top H g_0 g_0^\top H^2 g_0 = \left(\sum \alpha_i^2\right)\left(\sum \lambda_i^3 \alpha_i^2\right) - \left(\sum \lambda_i \alpha_i^2\right)\left(\sum \lambda_i^2 \alpha_i^2\right).$$

The nonnegativity follows from Chebyshev's sum inequality applied to the series $\{\lambda_i\}$ and $\{\lambda_i^2\}$.

8. Conclusion:

Since both terms are non-negative and the first is strictly positive, the inequality holds.

For sufficiently large $\rho_m$, the $\rho_m^2$ term dominates the Taylor expansion.

That is, $\exists \rho_0 > 0$ such that $\forall \rho_m > \rho_0$:

$$L\left(\vartheta_0 + \rho_m \frac{g_1}{\|g_1\|}\right) > L\left(\vartheta_0 + \rho_m \frac{g_0}{\|g_0\|}\right).$$

$\square$

**Remark.** *If $\rho_m$ is too small, the first-order term will dominate. The first-order term has*

$$\frac{g_0^\top g_1}{\|g_1\|} = \|g_0\| \cos(\phi) < \|g_0\|,$$

*where $\cos\phi = \frac{g_0^T g_1}{\|g_0\|\|g_1\|} < 1$ since $g_0$ and $g_1$ do not parallel. Thus, if $\rho_m$ is too small, it will have*

$$L\left(\vartheta_0 + \rho_m \frac{g_1}{\|g_1\|}\right) < L\left(\vartheta_0 + \rho_m \frac{g_0}{\|g_0\|}\right).$$

*From another perspective, this must hold because $g_0$ indicates the steepest ascent direction at $\vartheta_0$.*

**Remark.** *$\rho_m$ needs to be large only to ensure that the difference in the second-order term outweighs the first-order term, not intended to be too large to become impractical in real-world applications.*

### B.2 PROOF OF THE SECOND CONCLUSION

*Proof.*

1. Since $L$ admits a second-order approximation at $\theta_0$:

$$L\left(\vartheta_0 + \rho_m \frac{g_\alpha}{\|g_\alpha\|}\right) = L(\vartheta_0) + \rho_m \frac{g_0^\top g_\alpha}{\|g_\alpha\|} + \frac{\rho_m^2}{2} \frac{g_\alpha^\top H g_\alpha}{\|g_\alpha\|^2} + o(\rho_m^2),$$

$$L\left(\vartheta_0 + \rho_m \frac{g_1}{\|g_1\|}\right) = L(\vartheta_0) + \rho_m \frac{g_0^\top g_1}{\|g_1\|} + \frac{\rho_m^2}{2} \frac{g_1^\top H g_1}{\|g_1\|^2} + o(\rho_m^2).$$

2. Define the quadratic ratio:

$$f(\alpha) = \frac{g_\alpha^\top H g_\alpha}{\|g_\alpha\|^2}.$$

   At boundary points:

$$f(1) = \frac{g_1^\top H g_1}{\|g_1\|^2}, \quad f(0) = \frac{g_0^\top H g_0}{\|g_0\|^2}.$$

3. The derivative is:

$$f'(\alpha) = \frac{2(g_1 - g_0)^\top H g_\alpha \cdot \|g_\alpha\|^2 - 2(g_\alpha^\top H g_\alpha)(g_1 - g_0)^\top g_\alpha}{\|g_\alpha\|^4}.$$

   At $\alpha = 1$:

$$f'(1) = \frac{2}{\|g_1\|^4} \left[(g_1 - g_0)^\top H g_1 \cdot \|g_1\|^2 - (g_1^\top H g_1)(g_1 - g_0)^\top g_1\right].$$

4. Using $g_1 = g_0 + \rho H \frac{g_0}{\|g_0\|} + o(\rho)$:

$$g_1 - g_0 = \rho H \frac{g_0}{\|g_0\|} + o(\rho).$$

   Substituting into $f'(1)$:

$$f'(1) = \frac{2\rho}{\|g_1\|^4 \|g_0\|} \left[g_0^\top H^2 g_1 \cdot \|g_1\|^2 - (g_1^\top H g_1)(g_0^\top H g_1)\right] + o(\rho).$$

5. Further substituting $g_1 = g_0 + \rho H \frac{g_0}{\|g_0\|} + o(\rho)$ in:

$$\|g_1\|^2 = \left\|g_0 + \rho H \frac{g_0}{\|g_0\|} + o(\rho)\right\|^2 = \|g_0\|^2 + 2\rho \frac{g_0^\top H g_0}{\|g_0\|} + \rho^2 \frac{g_0^\top H^2 g_0}{\|g_0\|^2} + o(\rho^2),$$

$$g_0^\top H^2 g_1 \|g_1\|^2$$

$$= \left(g_0^\top H^2 g_0 + \rho \frac{g_0^\top H^3 g_0}{\|g_0\|} + o(\rho)\right)\left(\|g_0\|^2 + 2\rho \frac{g_0^\top H g_0}{\|g_0\|} + \rho^2 \frac{g_0^\top H^2 g_0}{\|g_0\|^2} + o(\rho^2)\right)$$

$$= g_0^\top H^2 g_0 \|g_0\|^2 + \rho \left(2 \frac{g_0^\top H^2 g_0 \cdot g_0^\top H g_0}{\|g_0\|} + \|g_0\| g_0^\top H^3 g_0\right)$$

$$+ \rho^2 \left(\frac{g_0^\top H^2 g_0 \cdot g_0^\top H^2 g_0}{\|g_0\|^2} + 2 \frac{g_0^\top H^3 g_0 \cdot g_0^\top H g_0}{\|g_0\|^2}\right) + o(\rho^2),$$

$$(g_1^\top H g_1)(g_0^\top H g_1)$$

$$= \left(g_0^\top H g_0 + 2\rho \frac{g_0^\top H^2 g_0}{\|g_0\|} + \rho^2 \frac{g_0^\top H^3 g_0}{\|g_0\|^2} + o(\rho^2)\right)\left(g_0^\top H g_0 + \rho \frac{g_0^\top H^2 g_0}{\|g_0\|} + o(\rho)\right)$$

$$= (g_0^\top H g_0)^2 + \rho \left(3 \frac{g_0^\top H g_0 \cdot g_0^\top H^2 g_0}{\|g_0\|}\right) + \rho^2 \left(\frac{g_0^\top H^3 g_0 \cdot g_0^\top H g_0}{\|g_0\|^2} + 2 \frac{(g_0^\top H^2 g_0)^2}{\|g_0\|^2}\right) + o(\rho^2).$$

6. Combining terms:

$$g_0^\top H^2 g_1 \|g_1\|^2 - (g_1^\top H g_1)(g_0^\top H g_1) = \left(g_0^\top H^2 g_0 \|g_0\|^2 - (g_0^\top H g_0)^2\right)$$
$$+ \rho \left(\|g_0\| g_0^\top H^3 g_0 - \frac{g_0^\top H g_0 \cdot g_0^\top H^2 g_0}{\|g_0\|}\right)$$
$$+ \rho^2 \left(\frac{g_0^\top H^3 g_0 \cdot g_0^\top H g_0 - (g_0^\top H^2 g_0)^2}{\|g_0\|^2}\right) + o(\rho^2).$$

7. Sign analysis:

- Zero-order term $g_0^\top H^2 g_0 \|g_0\|^2 - (g_0^\top H g_0)^2$: Strictly positive by Cauchy-Schwarz inequality since $H$ is positive definite and $g_0$ and $H g_0$ are not parallel.

- First-order term $\|g_0\|^2 g_0^\top H^3 g_0 - g_0^\top H g_0 \cdot g_0^\top H^2 g_0$: Non-negative by Chebyshev's sum inequality for the sequences $\{\lambda_i\}$ and $\{\lambda_i^2\}$ where $H = \sum \lambda_i v_i v_i^\top$.

- Second-order term $g_0^\top H^3 g_0 \cdot g_0^\top H g_0 - (g_0^\top H^2 g_0)^2$: Non-negative by Chebyshev's sum inequality.

8. Conclusion:

The term is strictly positive, which means $f'(1) > 0$. So, there exists $\alpha > 1$ such that $f(\alpha) > f(1)$. For sufficiently large $\rho_m$, where the second-order term dominates, this further implies:

$$L\left(\vartheta_0 + \rho_m \frac{g_\alpha}{\|g_\alpha\|}\right) > L\left(\vartheta_0 + \rho_m \frac{g_1}{\|g_1\|}\right).$$

□

**Remark.** *The $\rho_m$ threshold exists only to ensure the second-order term dominates the first-order term. In practice, moderate values suffice to observe XSAM's advantage over SAM.*

**Remark.** *Practically, the loss surface may not admit a second-order approximation, and the maximum does not necessarily lie around $\alpha = 1$. So we search a relatively large range of $\alpha$, e.g., in $[0, 2]$, to make it more generally applicable. Additionally, we use spherical linear combination instead, for a more uniform distribution of searched directions and better coverage.*

## C   COMPUTATIONAL OVERHEAD

The evaluation of each $\alpha$ will involve a forward pass of the neural network for calculating $L(\vartheta_0 + v(\alpha) \cdot \rho_m)$. So, the cost of the dynamic search of $\alpha^*$ roughly equals the number of samples of $\alpha$ times the cost of a forward pass. Typically, we use $20 \sim 40$ samples to search for $\alpha^*$. If this were required at every iteration, it would incur a considerable computational burden. Fortunately, frequent updates of $\alpha^*$ are unnecessary. According to our experiments, $\alpha^*$ is fairly stable and changes smoothly during training, as depicted in Figure 2 and Figure 13. In experiments, we by default adopt an epoch-wise update strategy: $\alpha^*$ is updated at the first iteration of each epoch and then kept fixed for the rest. Each epoch typically contains over 400 iterations. SAM requires $k + 1$ forward and $k + 1$ backward passes per iteration. So, the computational overhead of XSAM is roughly $40/(400 \cdot 2 \cdot (k + 1)) \le 0.025$, i.e., the increased cost is typically no more than $2.5\%$ when compared to SAM, which is negligible. A straightforward comparison of runtimes is presented in Table 1. The runtime of XSAM is nearly identical to that of SAM, indicating that the additional computational overhead is negligible.

## D   ADDITIONAL EXPERIMENTAL DETAILS FOR RESULTS IN SECTIONS 5

### D.1   DETAILS ABOUT THE 2D TEST FUNCTION

The test function used is defined by:

$$L(\theta) = L(\mu, \sigma) = -\log\left(0.7 e^{-K_1(\mu,\sigma)/1.8^2} + 0.3 e^{-K_2(\mu,\sigma)/1.2^2}\right), \tag{9}$$

where $K_i(\mu, \sigma)$ is the KL divergence between two univariate Gaussian distributions,

$$K_i(\mu, \sigma) = \log \frac{\sigma_i}{\sigma} + \frac{\sigma^2 + (\mu - \mu_i)^2}{2\sigma_i^2} - \frac{1}{2}. \tag{10}$$

with $(\mu_1, \sigma_1) = (20, 30)$ and $(\mu_2, \sigma_2) = (-20, 10)$. It features a sharp minimum at around $(-16.8, 12.8)$ with a value of $0.28$ and a flat minimum at around $(19.8, 29.9)$ with a value of $0.36$.

The visualized training trajectories in Figure 3a share the same start point $(-6.0, 10.0)$ and run for 400 steps. The learning rate is 5 (the gradient scale is small), momentum is 0.9, $\rho$ is 6.0, and $\rho_m$ is 18.0. The points passed at each step were recorded to plot the trajectories.

## D.2 Experiment Setup

**CIFAR-10, CIFAR-100, and Tint-ImageNet**. We use RandomCrop and CutOut (DeVries, 2017) augmentations for CIFAR-10 and CIFAR-100 while using RandomResizedCrop and RandomErasing (Zhong et al., 2020) augmentations for Tiny-ImageNet, since we believe improvements over strong augmentations can be more valuable. We use a batch size of 125 for all the datasets, such that the sample size of each dataset is divisible by the batch size, while near the typical choice of 128. We adopt the typical choice, SGD with a momentum of 0.9, as the base optimizer, which carries the true gradient descent to $\theta$. All models are trained for 200 epochs, while the cosine annealing learning rate schedule is adopted in all settings.

We run each experiment 5 times with different random seeds and calculate the mean and standard deviation. Each experiment was conducted using a single NVIDIA Tesla V100 GPU.

**ResNet50 on ImageNet**. We evaluate our method on the larger dataset, ImageNet. Standard data augmentation techniques are applied, including resizing, cropping, random horizontal flipping, and normalization. We take SGD as the base optimizer with a cosine learning rate decay.

**IWSLT2014**. We conduct experiments on the Neural Machine Translation (NMT) task, specifically German–English translation on the IWSLT2014 dataset (Cettolo et al., 2014), using the Transformer architecture following the FAIRSEQ (Ott et al., 2019). We use AdamW as the base optimizer due to its better performance on the transformer.

**ViT-Ti**. We further use a lightweight Vision Transformer (ViT-Ti) model on CIFAR-100 to evaluate our method. Note that following (Zhao et al., 2022a), we do not use Cutout augmentation for CIFAR-100 when trained by ViT-Ti. We use AdamW as the base optimizer.

## D.3 Hyperparameter Details

Table 5: Hyperparameter details for Results in Table 2.

| VGG-11 | CIFAR-10 | | | | CIFAR-100 | | | | Tiny-ImageNet | | | |
|---|---|---|---|---|---|---|---|---|---|---|---|---|
|  | SGD | SAM | WSAM | XSAM | SGD | SAM | WSAM | XSAM | SGD | SAM | WSAM | XSAM |
| Epoch | | 200 | | | | 200 | | | | 200 | | |
| Batch size | | 125 | | | | 125 | | | | 125 | | |
| Initial learning rate | | 0.05 | | | | 0.05 | | | | 0.05 | | |
| Momentum | | 0.9 | | | | 0.9 | | | | 0.9 | | |
| Weight decay | | $1 \times 10^{-3}$ | | | | $1 \times 10^{-3}$ | | | | $1 \times 10^{-3}$ | | |
| $\rho$ | – | 0.15 | 0.15 | 0.15 | – | 0.15 | 0.15 | 0.15 | – | 0.20 | 0.20 | 0.20 |
| $\rho_m$ | – | – | – | 0.30 | – | – | – | 0.30 | – | – | – | 1.20 |
| $\alpha$ | 0.0 | 1.0 | 0.75 | – | 0.0 | 1.0 | 1.0 | – | 0.0 | 1.0 | 1.0 | – |

| ResNet-18 | SGD | SAM | WSAM | XSAM | SGD | SAM | WSAM | XSAM | SGD | SAM | WSAM | XSAM |
|---|---|---|---|---|---|---|---|---|---|---|---|---|
| Epoch | | 200 | | | | 200 | | | | 200 | | |
| Batch size | | 125 | | | | 125 | | | | 125 | | |
| Initial learning rate | | 0.05 | | | | 0.05 | | | | 0.05 | | |
| Momentum | | 0.9 | | | | 0.9 | | | | 0.9 | | |
| Weight decay | | $1 \times 10^{-3}$ | | | | $1 \times 10^{-3}$ | | | | $1 \times 10^{-3}$ | | |
| $\rho$ | – | 0.15 | 0.15 | 0.15 | – | 0.15 | 0.15 | 0.15 | – | 0.20 | 0.20 | 0.20 |
| $\rho_m$ | – | – | – | 0.25 | – | – | – | 0.30 | – | – | – | 0.25 |
| $\alpha$ | 0.0 | 1.0 | 0.5 | – | 0.0 | 1.0 | 1.25 | – | 0.0 | 1.0 | 1.0 | – |

| DenseNet-121 | SGD | SAM | WSAM | XSAM | SGD | SAM | WSAM | XSAM | SGD | SAM | WSAM | XSAM |
|---|---|---|---|---|---|---|---|---|---|---|---|---|
| Epoch | | 200 | | | | 200 | | | | 200 | | |
| Batch size | | 125 | | | | 125 | | | | 125 | | |
| Initial learning rate | | 0.05 | | | | 0.05 | | | | 0.05 | | |
| Momentum | | 0.9 | | | | 0.9 | | | | 0.9 | | |
| Weight decay | | $1 \times 10^{-3}$ | | | | $1 \times 10^{-3}$ | | | | $1 \times 10^{-3}$ | | |
| $\rho$ | – | 0.05 | 0.05 | 0.05 | – | 0.10 | 0.10 | 0.10 | – | 0.20 | 0.20 | 0.20 |
| $\rho_m$ | – | – | – | 0.10 | – | – | – | 0.20 | – | – | – | 0.20 |
| $\alpha$ | 0.00 | 1.0 | 1.25 | – | 0.0 | 1.0 | 0.75 | – | 0.0 | 1.0 | 0.75 | – |

Table 6: Hyperparameter details for Results in Figure 3b. Note that, in this experiment, $\alpha$ for WSAM adopts the average value of the dynamic $\alpha^*$ in the corresponding XSAM. We see from the results that such WSAM already clearly outperforms SAM.

|  | $\rho$=0.10 | | | $\rho$=0.20 | | | $\rho$=0.30 | | |
|---|---|---|---|---|---|---|---|---|---|
|  | SAM | WSAM | XSAM | SAM | WSAM | XSAM | SAM | WSAM | XSAM |
| Epoch | | 200 | | | 200 | | | 200 | |
| Batch size | | 125 | | | 125 | | | 125 | |
| Initial learning rate | | 0.05 | | | 0.05 | | | 0.05 | |
| Momentum | | 0.9 | | | 0.9 | | | 0.9 | |
| Weight decay | | $1 \times 10^{-3}$ | | | $1 \times 10^{-3}$ | | | $1 \times 10^{-3}$ | |
| $\rho_m$ | – | – | 0.30 | – | – | 0.30 | – | – | 0.30 |
| $\alpha$ | 1.0 | 1.57 | – | 1.0 | 1.15 | – | 1.0 | 0.92 | – |

Table 7: Hyperparameter details for Results in Figure 4 and 14a. Note that the basic hyperparameters are provided here, while the other hyperparameters are clearly illustrated in the respective figures.

|  | Figure 4 | | Figure 14a | |
|---|---|---|---|---|
|  | SAM | XSAM | SAM | XSAM |
| Epoch | | 200 | | 200 |
| Batch size | | 125 | | 125 |
| Initial learning rate | | 0.05 | | 0.05 |
| Momentum | | 0.9 | | 0.9 |
| Weight decay | | $1 \times 10^{-3}$ | | $1 \times 10^{-3}$ |
| $\rho$ | | 0.15 | | 0.15 |

Table 8: Hyperparameter details for Results in Figure 3c. Note that, in this experiment, $\alpha$ for WSAM adopts the average value of the dynamic $\alpha^*$ in the corresponding XSAM. We see from the results that such WSAM already clearly outperforms SAM.

|  | $\rho$=0.04 | | | $\rho$=0.08 | | | $\rho$=0.12 | | |
|---|---|---|---|---|---|---|---|---|---|
|  | SAM | WSAM | XSAM | SAM | WSAM | XSAM | SAM | WSAM | XSAM |
| Epoch | | 200 | | | 200 | | | 200 | |
| Batch size | | 125 | | | 125 | | | 125 | |
| Initial learning rate | | 0.05 | | | 0.05 | | | 0.05 | |
| Momentum | | 0.9 | | | 0.9 | | | 0.9 | |
| Weight decay | | $1 \times 10^{-3}$ | | | $1 \times 10^{-3}$ | | | $1 \times 10^{-3}$ | |
| $\rho_m$ | – | – | 0.30 | – | – | 0.25 | – | – | 0.20 |
| $\alpha$ | 1.0 | 1.72 | – | 1.0 | 1.15 | – | 1.0 | 0.41 | – |

Table 9: Hyperparameters for SAM and XSAM on ImageNet/ResNet-50, Transformer/IWSLT2014, and ViT-Ti/CIFAR-100 in Table 3.

|  | ImageNet/ResNet-50 | | Transformer/IWSLT2014 | | CIFAR-100/ViT-Ti | |
|---|---|---|---|---|---|---|
|  | SAM | XSAM | SAM | XSAM | SAM | XSAM |
| Epoch | | 90 | | 300 | | 300 |
| Batch size / Max Token | | 512 | | 4096 | | 256 |
| Initial learning rate | | 0.2 | | $5 \times 10^{-4}$ | | 0.001 |
| Momentum | | 0.9 | | (0.9,0.98) | | (0.9,0.999) |
| Weight decay | | $1 \times 10^{-4}$ | | 0.3 | | 0.3 |
| Label smooth | | 0.0 | | 0.1 | | 0.1 |
| $\rho$ | | 0.05 | | 0.15 | | 0.9 |
| $\rho_m$ | – | 0.3 | – | 0.45 | – | 0.9 |
| $\alpha$ | 1.0 | – | 1.0 | – | 1.0 | – |

# E ADDITIONAL EXPERIMENTAL RESULTS AND ANALYSIS

## E.1 EVALUATION OF XSAM WITH OTHER SAM VARIANTS

In this section, we further evaluate the performance of SAM variants and their combinations with XSAM. As discussed in Section 4, some SAM variants, such as ASAM, FSAM, and VaSSO, target aspects of SAM that are largely orthogonal to those addressed by our method, making them potentially compatible for integration. Given the large number of such orthogonal approaches, we focus here on combining XSAM with ASAM and evaluating their performance on CIFAR-100 using ResNet-18. The results in Table 10 indicate that XSAM outperforms both SAM and ASAM individually. Furthermore, integrating XSAM with ASAM leads to further improvement, demonstrating the effectiveness of XSAM in combination with other SAM variants.

Table 10: Test accuracy of SAM variants and their combinations with XSAM.

|  | SAM | ASAM | XSAM | XSAM+ASAM |
|---|---|---|---|---|
| Test Accuracy | $80.93 \pm 0.11$ | $81.11 \pm 0.06$ | $81.24 \pm 0.07$ | $\mathbf{81.68 \pm 0.11}$ |

We have additionally compared XSAM with ASAM, VaSSO, and WSAM on CIFAR-100 using both ResNet-18 and DenseNet-121. As shown in the Table 11, XSAM achieves the highest accuracy across both architectures, further demonstrating its effectiveness.

Table 11: Comparison on CIFAR-100 with ResNet-18 and DenseNet-121. All baseline methods are carefully tuned for optimal performance. XSAM uses the same $\rho$ as SAM, as in the paper.

| Method | ResNet-18 | DenseNet-121 |
|---|---|---|
| SAM | $80.93 \pm 0.11$ | $83.81 \pm 0.02$ |
| ASAM | $81.11 \pm 0.06$ | $83.99 \pm 0.25$ |
| VaSSO | $80.84 \pm 0.15$ | $83.78 \pm 0.25$ |
| WSAM | $80.95 \pm 0.19$ | $83.91 \pm 0.15$ |
| XSAM | $81.24 \pm 0.07$ | $83.96 \pm 0.10$ |
| XSAM + ASAM | $\mathbf{81.68 \pm 0.11}$ | $\mathbf{84.06 \pm 0.21}$ |

## E.2 ADDITIONAL EXPERIMENTS OF MULTI-STEP SAM

We additionally compare multi-step SAM variants and XSAM under varying $\rho$. As we see in Table 12, all of these variants, especially LSAM and MSAM+, which involve intermediate gradients rather than merely using the last gradient $g_k$, managed to get consistently superior results than SAM. The performance of SAM constantly decreases as $\rho$ gets large, which, from our perspective, suggests the deviations of $g_k$ from $\vartheta_0$ are too large. Under such circumstances, the earlier $g_i$ must have less deviation, so combining it with earlier gradients would help. Besides, we see no clear trend for LSAM, LSAM+, MSAM, and MSAM+ as $\rho$ gets large.

Although MSAM+ can be viewed as LSAM+ with weights of gradients changed from $1/\|g_i\|$ to simply 1, the performance gap between them is obvious. This demonstrates that the weighting of gradients at different steps affects performance, and a more appropriate weighting scheme can lead to higher accuracy. Regardless, XSAM consistently outperforms all these methods in all cases.

We further compare XSAM with MSAM and LSAM under $k = 1, 2, 4$ on DenseNet-121 using CIFAR-100 and on ResNet-18 using CIFAR-10. As shown in Tables 13 and 14, XSAM consistently attains high accuracy while maintaining strong robustness. In contrast, existing multi-step SAM variants may even underperform their single-step counterparts.

Table 12: Results on CIFAR-100 using ResNet-18 in multi-step ($k = 3$) setting.

| Method | $\rho = 0.04$ | $\rho = 0.08$ | $\rho = 0.12$ |
|---|---|---|---|
| SAM | $80.79_{\pm 0.41}$ | $80.75_{\pm 0.27}$ | $79.72_{\pm 0.33}$ |
| LSAM | $81.00_{\pm 0.21}$ | $81.20_{\pm 0.24}$ | $81.16_{\pm 0.04}$ |
| LSAM+ | $80.56_{\pm 0.20}$ | $80.77_{\pm 0.04}$ | $80.21_{\pm 0.27}$ |
| MSAM | $81.04_{\pm 0.06}$ | $81.12_{\pm 0.17}$ | $80.93_{\pm 0.11}$ |
| MSAM+ | $80.72_{\pm 0.16}$ | $81.16_{\pm 0.05}$ | $81.16_{\pm 0.05}$ |
| XSAM | $\mathbf{81.23}_{\pm 0.06}$ | $\mathbf{81.36}_{\pm 0.08}$ | $\mathbf{81.29}_{\pm 0.06}$ |

Table 13: Results on DenseNet-121 with CIFAR-100 with different $k$. $\rho = \rho^*/k$ with $\rho^*$ for single-step.

| Method | $k = 1$ | $k = 2$ | $k = 4$ |
|---|---|---|---|
| LSAM | $83.81 \pm 0.02$ | $83.82 \pm 0.28$ | $83.40 \pm 0.17$ |
| MSAM | $83.81 \pm 0.02$ | $83.67 \pm 0.23$ | $83.74 \pm 0.18$ |
| XSAM | $\mathbf{83.96 \pm 0.10}$ | $\mathbf{84.05 \pm 0.04}$ | $\mathbf{84.02 \pm 0.31}$ |

Table 14: Results on ResNet-18 with CIFAR-10 with different $k$. $\rho = \rho^*/k$ with $\rho^*$ for single-step.

| Method | $k = 1$ | $k = 2$ | $k = 4$ |
|---|---|---|---|
| LSAM | $96.59 \pm 0.06$ | $96.66 \pm 0.03$ | $96.72 \pm 0.07$ |
| MSAM | $96.59 \pm 0.06$ | $96.78 \pm 0.05$ | $96.80 \pm 0.07$ |
| XSAM | $\mathbf{96.74 \pm 0.04}$ | $\mathbf{96.81 \pm 0.06}$ | $\mathbf{96.81 \pm 0.11}$ |

### E.3 ADDITIONAL EXPERIMENTS ON CORRUPTED DATASETS

We have conducted additional experiments on CIFAR-10-C and CIFAR-100-C using ResNet-18 and DenseNet-121. Specifically, we consider 19 types of corruptions, each applied at five severity levels, and group them into four categories: noise, blur, weather, and digital.

We report the mean accuracy as the evaluation metric, with higher values indicating better performance. The results, as shown in the Table 15, indicate that XSAM consistently achieves high performance and demonstrates robustness across all settings.

Table 15: Performance on CIFAR-10-C/CIFAR-100-C with ResNet-18 and DenseNet-121.

(a) CIFAR-100-C, ResNet-18

| Method | Noise | Blur | Weather | Digital | Overall |
|---|---|---|---|---|---|
| SGD | 22.36 | 47.47 | 55.44 | 60.03 | 48.07 |
| SAM | 25.58 | 51.14 | 58.82 | 63.30 | 51.45 |
| XSAM | 25.44 | 52.65 | 59.83 | 63.54 | **52.07** |

(b) CIFAR-10-C, ResNet-18

| Method | Noise | Blur | Weather | Digital | Overall |
|---|---|---|---|---|---|
| SGD | 51.14 | 71.85 | 85.23 | 85.85 | 74.76 |
| SAM | 53.65 | 75.91 | 85.23 | 86.48 | 76.59 |
| XSAM | 55.13 | 75.94 | 85.76 | 85.99 | **76.81** |

(c) CIFAR-100-C, DenseNet-121

| Method | Noise | Blur | Weather | Digital | Overall |
|---|---|---|---|---|---|
| SGD | 26.07 | 51.12 | 59.93 | 63.78 | 51.90 |
| SAM | 29.78 | 55.22 | 63.58 | 67.26 | 55.62 |
| XSAM | 31.02 | 56.73 | 64.15 | 67.25 | **56.37** |

(d) CIFAR-10-C, DenseNet-121

| Method | Noise | Blur | Weather | Digital | Overall |
|---|---|---|---|---|---|
| SGD | 49.50 | 73.51 | 85.17 | 85.88 | 74.85 |
| SAM | 54.75 | 77.52 | 87.30 | 87.85 | 78.08 |
| XSAM | 55.94 | 77.05 | 87.60 | 87.72 | **78.20** |

### E.4 INNER PROPERTIES OF XSAM

In this section, we present investigations into the internal properties of XSAM.

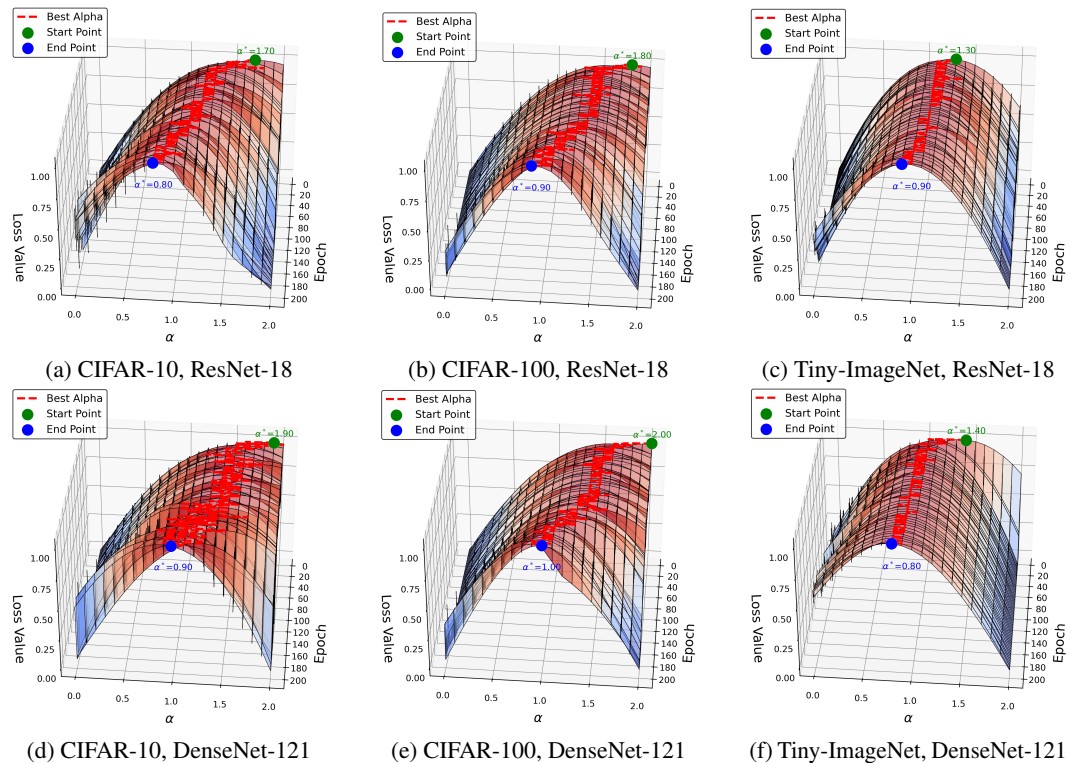

Figure 13: More visualizations of the dynamic estimations of $\alpha$.

We first visualize the dynamic evaluations of $\alpha$ in a training instance in Figure 2 and in Figure 13, where loss values are normalized for better visibility. As we can see, for every dynamic evaluation of $\alpha$, there is a clear optimal $\alpha$. With the epoch-wise evaluation of $\alpha$, we still see that the change of $\alpha^*$ during training is very smooth, which supports our choice of less frequently updating $\alpha^*$ for reducing computational overhead. On the other hand, we do see that $\alpha^*$ is changing during training, which validates our argument that a fixed $\alpha$ may not be optimum.

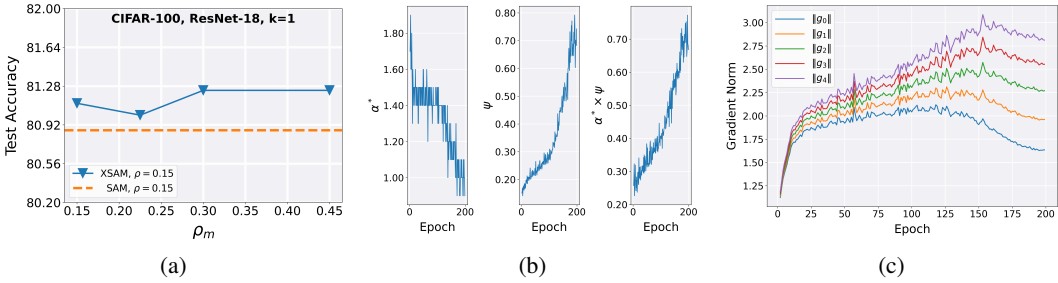

Figure 14: (a) Robustness analysis of XSAM with respect to $\rho_m$. (b)Training statistics of XSAM. (c) The norms of $g_i$ during training.

We further study how $\rho_m$ influences the final performance. As results presented in Figure 14a, while $\rho_m$ does impact performance to some extent, XSAM is able to outperform SAM in a fairly large range of $\rho_m$, from $\rho$ to $3\rho$. So, we consider that XSAM is not sensitive to $\rho_m$. The counterpart, as to how $\rho$ influences when fixing the $\rho_m$, is actually demonstrated in Figure 3b, where we have used a fixed $\rho_m = 0.3$ by intention. It seems fairly robust to $\rho$.

In our experiments, we also see that $\alpha^*$ has a decreasing tendency during training. In fact, the angle $\psi$ between $v_0$ and $v_1$ has an increasing tendency during training. We visualize such changes along with the offset angle $\alpha^* \cdot \psi$ from $v_0$ to the direction of the local maximum in Figure 14b. We see that

the offset angle $\alpha^* \cdot \psi$ tends to increase. This may be because it converges to a lower position in a minima region as the learning rate decreases. Nevertheless, XSAM is able to help it away from the maximum within the local neighborhood in any case, as evident by the test accuracy results.

We show in Figure 14c an instance of norm change of $g_i$ during training in multi-step settings.

### E.5 The Flatness/Sharpness of Resulting Models

**Hessian spectrum**. To demonstrate that XSAM converges to flatter minima (more precisely, successfully shifts to a region where the maximum within the local neighborhood is lower), we calculate the Hessian eigenvalues of ResNet-18 trained for 200 epochs on CIFAR-10 with SGD, SAM, and XSAM. Following (Foret et al., 2020; Jastrzebski et al., 2020; Mi et al., 2022), we adopt two metrics: the largest eigenvalue (i.e., $\lambda_1$) and the ratio of the largest eigenvalue to the fifth largest one (i.e., $\lambda_1/\lambda_5$). To avoid the expensive computation cost of exact Hessian spectrum calculation, we approximate eigenvalues using the Lanczos algorithm (Ghorbani et al., 2019). The results, shown in **Table 16**, indicate that XSAM yields the smallest hessian spectrum, suggesting that it converges to flatter minima than SAM and SGD.

Table 16: Hessian spectrum of ResNet-18 using SGD, SAM, and XSAM on CIFAR-10.

|  | SGD | SAM | XSAM |
|---|---|---|---|
| $\lambda_1$ | 78.79 | 36.15 | 33.92 |
| $\lambda_1/\lambda_5$ | 2.26 | 1.89 | 1.59 |

**Visualization of loss landscape**. We visualize the loss landscape of ResNet-18 trained on CIFAR-10 with SGD, SAM, and XSAM to further compare the flatness of the minimum. Using the visualization procedure in (Li et al., 2018), we randomly choose orthogonal normalization directions (i.e., X axis and Y axis) and then sample $50 \times 50$ points in the range of [-1,1] from these two directions. As shown in Figure 15, XSAM has a flatter loss landscape than SAM and SGD.

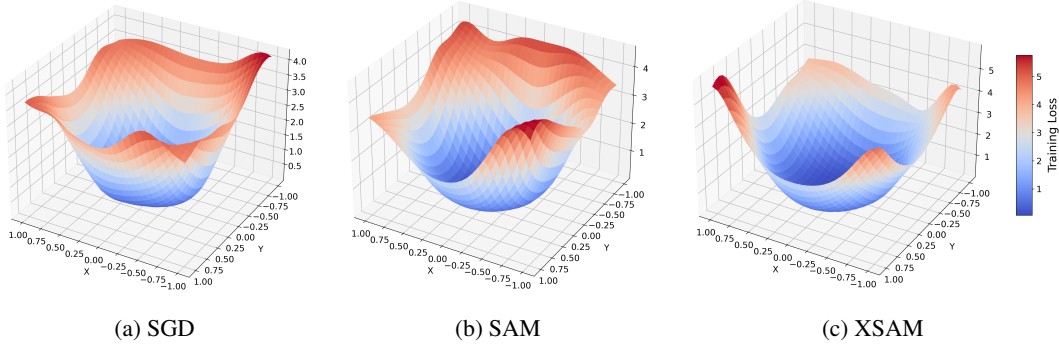

(a) SGD          (b) SAM          (c) XSAM

Figure 15: Loss landscape visualizations of ResNet-18 on CIFAR-10 with SGD, SAM, and XSAM.

**Average sharpness**. We further visualize the average sharpness of the loss landscape at the convergence point. Specifically, following (Foret et al., 2020), we define the sharpness as the difference between the loss of the perturbation point and the loss of the convergence point. The average sharpness is then computed as the mean sharpness over multiple perturbations under the same perturbation radius. Then, we sample multiple random directions (e.g., 10, 50, 250, 1250) and continue this process until the average sharpness loss curve stabilizes, which provides a more representative characterization of the loss behavior around the convergence point. Based on our experiments, sampling 250 random directions is sufficient to achieve stable results. In addition, for the perturbation method, we adopt filter-wise and element-wise perturbation following (Li et al., 2018) to ensure a fair comparison between different optimizers (i.e., SGD, SAM, and XSAM). As shown in Figure 16, SAM exhibits smaller average sharpness compared to SGD, while XSAM further reduces the average sharpness.

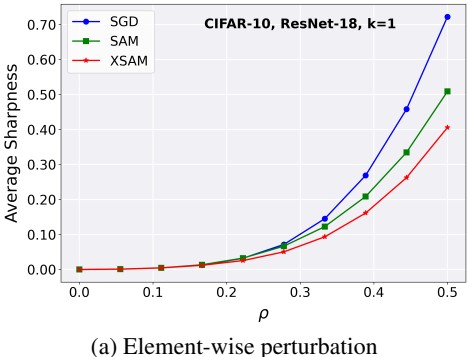 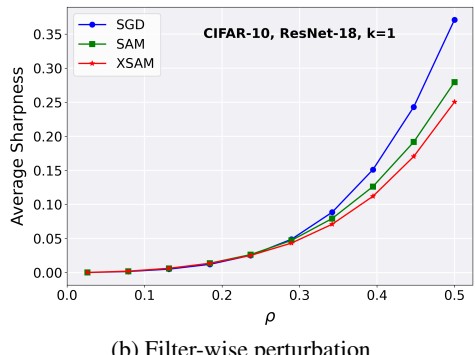

(a) Element-wise perturbation                    (b) Filter-wise perturbation

Figure 16: Visualization of the average sharpness of the loss landscape at the convergence point.

## F  STRATEGIES FOR GRADIENT SCALE

Our default gradient scale strategy is using $\|g_k\|$ to match the scale with SAM. In this section, we empirically study a set of different ways for setting the gradient scale, which includes: typical choices like $\|g_k\|$ and $\|g_0\|$, simple extensions like $\sum_{i=0}^{k} \|g_i\|/(k+1)$ and $\max_{i=0}^{k} \|g_i\|$. Besides, we further explored two slope-based strategies:

$$
\text{slope}_k := \frac{L(\vartheta_k) - L(\vartheta_0)}{\|\vartheta_k - \vartheta_0\|},
$$
$$
\text{slope}_m := \frac{L(\vartheta_0 + v(\alpha) \cdot \rho_m) - L(\vartheta_0)}{\rho_m},
$$

which is the averaged slope from $\vartheta_0$ to $\vartheta_k$ and from $\vartheta_0$ to the approximated maximum, respectively.

Note that since our direction is away from the approximated maximum, it can be an interesting combination when using the slope from $\vartheta_0$ to the approximated maximum as the gradient scale, which shares the same intrinsic core as stochastic gradient descent. However, it would require an extra forward pass to evaluate $L(\vartheta_0 + v(\alpha) \cdot \rho_m)$.

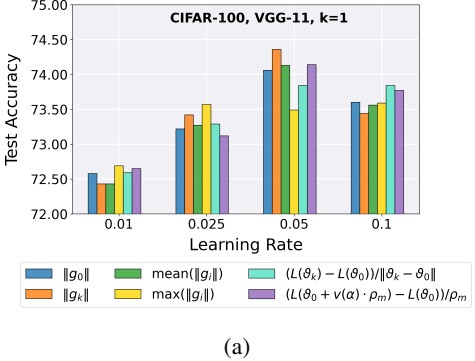 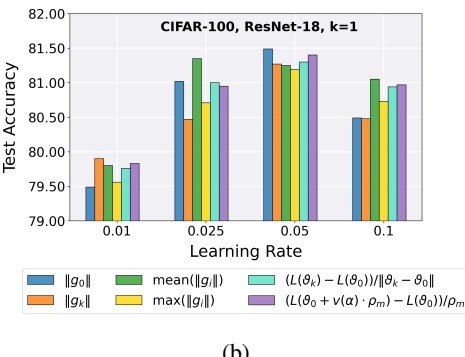

(a)                                          (b)

Figure 17: Comparison of various gradient scale strategies.

The results are shown in Figure 17. As we can see, the gradient scale seems to be something that is even more mysterious than the gradient direction. It is hard to draw a direct conclusion on which might be the best choice among such a reasonably large group. Nevertheless, some choices appear to be good in most circumstances, which may include $\|g_0\|$, $\|g_k\|$, and slope$_m$. These primary results are included for completeness. Notably, the work (Tan et al., 2025) argues that rescaling the gradient using $\|g_0\|$ is more stable than using $\|g_1\|$. In our experiments, however, we do not observe a noticeable stability advantage. We would leave further investigation into this as future work.

## G  ADDITIONAL RELATED WORK

The connection between flatness/sharpness and generalization was realized early on (Hochreiter & Schmidhuber, 1994) and further explored in subsequent works (Hochreiter & Schmidhuber, 1997; McAllester, 1999; Neyshabur et al., 2017; Jiang et al., 2019), motivating efforts toward finding flatter solutions. While SGD is believed to favor flat minima implicitly (Keskar et al., 2016; Ma & Ying, 2021), more explicit methods are preferred and developed. Typical instances include Entropy-SGD (Chaudhari et al., 2017) that employs entropy regularization, SWA (Izmailov et al., 2018) that seeks flatness by averaging model parameters, and SAM (Foret et al., 2020) that optimizes sharpness.

There are some variants that focus on improving the performance of multi-step SAM. Vanilla multi-step SAM (Foret et al., 2020) updates the model using the gradient at the last step. MSAM (Kim et al., 2023) suggests averaging all gradients except the first gradient at the original location. Lookbehind-SAM (LSAM) (Mordido et al., 2024) suggests another way that utilizes all gradients but excludes the first. In comparison, in multi-step settings, our method leverages all gradients ($\{g_i\}_{i=0}^{k-1}$ in $v_0$, and $g_k$ in $v_1$) in a dynamic interpolation manner and explicitly approximates the direction of the maximum.

There are also some works that seek to reduce the computational overhead of SAM. For instance, ESAM (Du et al., 2021) achieves this via stochastic weight perturbation and sharpness-sensitive data selection. SSAM (Mi et al., 2022) accelerates SAM with a sparse perturbation. LookSAM (Liu et al., 2022a) reduces computational overhead by computing SAM's gradient only periodically and relying on an approximate gradient for most of the training time. RST (Zhao et al., 2022b) and AE-SAM (Jiang et al., 2023) suggest alternating between SGD and SAM in randomized and adaptive ways, respectively.

Another important line of research on SAM focuses on understanding its underlying mechanism. For instance, (Wen et al., 2023) finds that the gradient of SAM aligns with the top eigenvector of the Hessian in the late phase of training. This phenomenon is also concurrently found by (Bartlett et al., 2023). (Andriushchenko et al., 2023a) argues that SAM leads to low-rank features. In addition, an interesting fact observed by (Andriushchenko & Flammarion, 2022) is that training with SAM only in the late phase of training can achieve an improvement similar to that of full training with SAM. A recent work (Zhou et al., 2025) further analyzes and theoretically shows the learning dynamics of applying SAM late in training. (Tahmasebi et al., 2024) introduces a universal class of sharpness measures, in which SAM, known for its bias toward minimizing the maximum eigenvalue of the Hessian matrix, can be regarded as a special case. Our work is orthogonal to these works, providing a new perspective for understanding a fundamental question of why applying the gradient from the ascent point to the current parameters is valid. At the same time, we propose XSAM as a better alternative.

In addition, SAM achieves extraordinary performance on various tasks. For instance, it has proven particularly effective in long-tail learning (Rangwani et al., 2022b). ImbSAM (Zhou et al., 2023) applies SAM only to the tail classes to improve their generalization. Further, CC-SAM (Gowda & Clifton, 2024) generates class-specific perturbations for each class, although this comes at an increased computational cost. Focal-SAM (Li et al., 2025) aims to achieve fine-grained sharpness control for each class while maintaining efficiency. These SAM variants are specialized in long-tail learning and differ from our work.

## H  USE OF LARGE LANGUAGE MODELS

We used a large language model (LLM) only for language polishing (grammar, wording, and clarity) of drafts written by the authors. The model did not generate research ideas, methods, analyses, results, or figures, and it did not write any sections from scratch.

