# OpenReview forum: "Revisiting Sharpness-Aware Minimization: A More Faithful and Effective Implementation"
_ICLR.cc/2026/Conference — ICLR 2026 Poster_

### Official Review · Reviewer_SZGB · 2025-10-19

**Soundness:** 2
**Presentation:** 3
**Contribution:** 2
**Rating:** 4
**Confidence:** 4

**Summary:**

This paper proposes a new variant of SAM optimizer, namely, XSAM that provides a more faithful and effective estimation of the gradient with respect to the current parameters. The authors argue that the gradient at the single-step ascent point offers a better approximation of the direction from the current parameters towards the maximum within the local neighborhood than the local gradient. And they further justify this observation under a second-order approximation, theoretically. Extensive experiments are also presented to validate the efficacy of the proposed optimizer.

**Strengths:**

This paper is well-motivated and easy to follow. Actually, SAM optimizer has attracted a lot of attention and many works have been devoted to improve the generalization performance of SAM. In this work, the authors identified a critical issue in estimating the gradient descent direction and proposes a variant of SAM that unifies one-step and multi-step versions.

**Weaknesses:**

While the authors have presented their work in a good shape, I still have serveral questions:

 - In line 22, the authors claim that **thereby enabling a more direct escape from the maximum within the local neighborhood**. Actually, I do not catch why is this case. As Figure 1a suggests, descending along $g_0$ quickly steers the optimization process towards the low-loss regions,  then $g_1@\theta_0$,  and $g_1$ gradually mitigates this issue. Frankly speaking, I do not know how it relates to the maximum of the local neighborhood.

 - In line 72, the authors states that **...eveal that the approximation by the gradient at the single-step ascent point is often inaccurate**. Possibly, here might be a mistake. Please recall that Figure 1a is visualized by spanning $g_0$ and $g_1$. And, the direction of $g_1$ is heavily dependent on the perturation radius $\rho$.  Therefore, the loss peak might vary significantly. It would be much better to visulize the true gradient with respect to the current parameters as well. Moreover, what's the value of $\rho$?

 - On the theoretical respect, my biggest concern is that the proof of Proposition 1 is too hand-wavy. In line 810,  **for sufficiently large $\rho_m$, the $\rho_m^2$ term dominates**. I believe here deserves further investigation because the taylor-expansion holds only for small $\rho_m$ and the remainding terms cannot be simply removed. Moreover, the assumption that Hessian $H$ is positive definite also requires careful handling and experimental verification. Idealy, I would suggest to plot $L(\theta_0+\rho_m\frac{g_1}{\|g_1\|})$ versus $L(\theta_0+\rho_m\frac{g_0}{\|g_0\|})$ for different combinations of $\rho$ and $\rho_m$.

- In Experiments section, more baselines should be included, such as ASAM, FisherSAM, etc. Particularly, line 16 of Algorithm 1 rescales the descent direction with $\|g_k\|$. I am wondering how it performs when rescaled with the base gradient $\|g_0\|$? The result of a recent study [1] seems to suggest it works. Moreover, how XSAM performs on corrupted dataset such as CIFAR-10/100-C should be reported as well.

[1] Tan et al., Stabilizing sharpness-aware minimization through a simple renormalization strategy, JMLR, 2025.

**Questions:**

please see Weaknesses.

---

> ### Author Response · Authors · 2025-11-24
> **Response to Reviewer SZGB, Part 1/3**
>
> Thanks for your time and effort in reviewing our work. We sincerely appreciate your recognition of the motivation and significance of the identified issue in our work. We address the questions and concerns you raised as follows.
>
> **W1-1: In line 22, the authors claim that "thereby enabling a more direct escape from the maximum within the local neighborhood". Why is this the case?**
>
> - Note that by "the maximum", we mean "the point with the maximum loss".
>
> - Updating the parameters in the direction opposite to the direction toward the maximum will naturally drive the parameters away from that maximum.
>
> - $g_1$, when applied to $\vartheta_0$, better approximates the direction toward the maximum (than $g_0$). So, updating the parameters $\vartheta_0$ along the opposite direction of $g_1$ would move the parameters more directly away from the maximum (than along the opposite direction of $g_0$).
>
> **W1-2: As Figure 1(a) suggests, descending along $g_0$ quickly steers the parameters toward the low-loss regions, then, $g_1@\vartheta_0$ and $g_1$ gradually mitigate this issue. I do not know how it relates to the maximum of the local neighborhood.**
>
> - Moving the parameters away from the maximum within the local neighborhood will directly minimize the maximum loss within the local neighborhood. This is just like moving parameters in the opposite direction of the local gradient (which can be viewed as moving away from the linearly approximated high-loss region) will directly minimize the local loss.
>
> - Descending along the local gradient $g_0$ will only minimize the local loss. SAM aims to minimize the maximum loss within the local neighborhood. So, we need to descend along the direction toward the maximum.
>
> - Our point is not that $g_1@\vartheta_0$ is mitigating the quick convergence toward the low-loss region. Our point is that it better approximates the direction toward the maximum. So, descending along it will more effectively minimize the maximum loss within the local neighborhood.
>
> - Descending along $g_1@\vartheta_0$ may not reduce the local loss as quickly or as effectively as the local gradient. Regardless, minimizing the local loss is not our goal. Our goal is to minimize the maximum loss within the local neighborhood, which it more effectively achieves.
>
> **W2-1: In line 72, the authors state that "reveal that the approximation by the gradient at the single-step ascent point is often inaccurate". Here might be a mistake. Recall that Figure 1(a) is visualized by spanning $g_0$ and $g_1$. And the direction of $g_1$ is heavily dependent on the perturbation radius $\rho$. Therefore, the loss peak might vary significantly.**
>
> - You are right that the loss peak in the visualized 2D hyperplane may vary, since one of its spanning vectors, $g_1$, depends on the perturbation radius.
>
> - We implicitly assume that the loss surface is smooth in the high-dimensional space, and that the direction of the high-loss region remains stable across different 2D slices of the loss landscape.
>
> - We believe that our assumption is reasonable. Note that even if it is considered to weaken the argument by visualization, our claim that "the approximation by the gradient at the single-step ascent point is often inaccurate" remains valid, as it is further justified by two additional arguments presented in the paper.
>     - The direction of $g_1$ is highly sensitive to the perturbation radius $\rho$. Even if the approximation is accurate for some $\rho$, given the sensitivity, it is likely to be inaccurate for the others. Furthermore, as $\rho$ approaches zero, $g_1$ will reduce to $g_0$, and its quality will become identical to $g_0$. So, it must often be inaccurate.
>     - The second statement of Proposition 1 shows that there generally exists a better direction than $g_1$, which means its approximation (under the assumption) is always inaccurate.
>
> **W2-2: It would be much better to visualize the true gradient with respect to the current parameters as well.**
>
> - $g_0$ is exactly the true gradient with respect to the current parameters, which has already been visualized.
>
> **W2-3: What's the value of $\rho$?**
>
> - The $\rho$ for Figure 1(a) is $0.15$. The training is on CIFAR-100 using ResNet-18.

---

> > ### Author Response · Authors · 2025-11-24
> > **Response to Reviewer SZGB, Part 2/3**
> >
> > **W3-1: In line 810, for sufficiently large $\rho_m$, the $\rho_m^2$ term dominates. I believe this deserves further investigation because the Taylor expansion holds only for small $\rho_m$ and the remaining terms cannot be simply removed.**
> >
> > - We assume that the loss admits a second-order approximation. Or in other words, it is *essentially quadratic*. So, its second-order Taylor expansion holds for arbitrarily large $\rho_m$. The remaining terms are, by assumption, negligible.
> >
> > - It is worth noting that this assumption is merely to simplify our establishment that the second-order term dominates the first-order term, which may hold under more general conditions.
> >
> > **W3-2: The assumption that the Hessian is positive definite also requires careful handling and experimental verification.**
> >
> > - We acknowledge that the assumptions, including a positive definite Hessian, are idealized. Regardless, the purpose of Proposition 1 is solely to theoretically affirm two claims: SAM better approximates the direction toward the maximum than SGD, and the approximation by SAM is typically suboptimal. Importantly, it demonstrates that both may even hold in the quadratic case.
> >
> > **W3-3: I would suggest to plot $L(\vartheta_0 + \rho_m g_1 / ||g_1||)$ versus $L(\vartheta_0 + \rho_m g_0 / ||g_0||)$ for different combinations of $\rho$ and $\rho_m$.**
> >
> > - Thank you for the suggestion. Following your suggestion, we plot $L(\vartheta_0 + \rho_m g_1 / ||g_1||)$ and $L(\vartheta_0 + \rho_m g_0 / ||g_0||)$ across various $(\rho, \rho_m)$ combinations. As shown in Figures 11 and 12 in Appendix A of the revised manuscript, when $\rho_m$ is larger than a moderate threshold, the loss along $g_1$ indeed exceeds that along $g_0$.

---

> > > ### Author Response · Authors · 2025-11-24
> > > **Response to Reviewer SZGB, Part 3/3**
> > >
> > > **W4-1: In experiments, more baselines should be included, such as ASAM, FisherSAM, etc.**
> > >
> > > Thank you for your suggestion.
> > >
> > > - We have additionally compared XSAM with ASAM, VaSSO, and WSAM on CIFAR-100 using both ResNet-18 and DenseNet-121. As shown in the table below, XSAM achieves the highest accuracy across both architectures, further demonstrating its effectiveness. The results and related details have been incorporated into the revised manuscript in Appendix E.1 (highlighted in blue).
> > >
> > > - We note that the techniques in some baselines, e.g., ASAM and VaSSO, are orthogonal to our work, and could potentially be combined with XSAM for further improvements.
> > >
> > > - CIFAR-100 on ResNet-18 and DenseNet-121:
> > > | | | |
> > > |:---:|:---:|:---:|
> > > | Method | ResNet-18 | DenseNet-121 |
> > > | SAM | 80.93 $\pm$ 0.11 | 83.81 $\pm$ 0.02 |
> > > | ASAM | 81.11 $\pm$ 0.06 | 83.99 $\pm$ 0.25 |
> > > | VaSSO | 80.84 $\pm$ 0.15 | 83.78 $\pm$ 0.25 |
> > > | WSAM | 80.95 $\pm$ 0.19 | 83.91 $\pm$ 0.15 |
> > > | XSAM | 81.24 $\pm$ 0.07 | 83.96 $\pm$ 0.10 |
> > > | XSAM + ASAM | **81.68 $\pm$ 0.11** | **84.06 $\pm$ 0.21** |
> > >
> > >     #### All baseline methods are carefully tuned for optimal performance. XSAM uses the same $\rho$ as SAM, as in the paper.
> > >
> > > **W4-2: Line 16 of Algorithm 1 rescales the descent direction with $|g_k|$. I am wondering how it performs when rescaled with the base gradient $|g_0|$? The result of a recent study [1] seems to suggest it works.**
> > >
> > > - In Appendix F, we have studied and compared different scaling strategies, including $|g_0|$, $|g_1|$, and four more complex strategies. We do not obtain a consistent conclusion regarding which strategy is definitively better: the effectiveness of these strategies varies across models and settings.
> > >
> > > - The study [1] argues that rescaling the gradient using $|g_0|$ is more stable than using $|g_1|$. In our experiments, however, we do not observe a significant stability advantage. We have included a discussion of [1] in Appendix F (highlighted in blue).
> > >
> > > [1] Tan et al., Stabilizing sharpness-aware minimization through a simple renormalization strategy, JMLR, 2025.
> > >
> > > **W4-3: How XSAM performs on corrupted dataset such as CIFAR-10/100-C should be reported as well.**
> > >
> > > - Thank you for the suggestion. Following your suggestion, we have conducted additional experiments on CIFAR-10-C and CIFAR-100-C using ResNet-18 and DenseNet-121. Specifically, we consider 19 types of corruptions, each applied at five severity levels, and group them into four categories: noise, blur, weather, and digital. We report the mean accuracy as the evaluation metric, with higher values indicating better performance.
> > >
> > > - The results, as shown in the tables below, indicate that XSAM consistently achieves high performance and demonstrates robustness across all settings. These results have been incorporated into the revised manuscript in Appendix E.3 (highlighted in blue).
> > >
> > > - CIFAR-100-C, ResNet-18:
> > > | | | | | | |
> > > |:---:|:---:|:---:|:---:|:---:|:---:|
> > > | Method | Noise | Blur | Weather | Digital | Overall |
> > > | SGD | 22.36 | 47.47 | 55.44 | 60.03 | 48.07 |
> > > | SAM | 25.58 | 51.14 | 58.82 | 63.30 | 51.45 |
> > > | XSAM | 25.44 | 52.65 | 59.83 | 63.54 | **52.07** |
> > >
> > > - CIFAR-10-C, ResNet-18:
> > > | | | | | | |
> > > |:---:|:---:|:---:|:---:|:---:|:---:|
> > > | Method | Noise | Blur | Weather | Digital | Overall |
> > > | SGD | 51.14 | 71.85 | 85.23 | 85.85 | 74.76 |
> > > | SAM | 53.65 | 75.91 | 85.23 | 86.48 | 76.59 |
> > > | XSAM | 55.13 | 75.94 | 85.76 | 85.99 | **76.81** |
> > >
> > > - CIFAR-100-C, DenseNet-121:
> > > | | | | | | |
> > > |:---:|:---:|:---:|:---:|:---:|:---:|
> > > | Method | Noise | Blur | Weather | Digital | Overall |
> > > | SGD | 26.07 | 51.12 | 59.93 | 63.78 | 51.90 |
> > > | SAM | 29.78 | 55.22 | 63.58 | 67.26 | 55.62 |
> > > | XSAM | 31.02 | 56.73 | 64.15 | 67.25 | **56.37** |
> > >
> > > - CIFAR-10-C, DenseNet-121:
> > > | | | | | | |
> > > |:---:|:---:|:---:|:---:|:---:|:---:|
> > > | Method | Noise | Blur | Weather | Digital | Overall |
> > > | SGD | 49.50 | 73.51 | 85.17 | 85.88 | 74.85 |
> > > | SAM | 54.75 | 77.52 | 87.30 | 87.85 | 78.08 |
> > > | XSAM | 55.94 | 77.05 | 87.60 | 87.72 | **78.20** |
> > >
> > > ---
> > > We sincerely hope that the above clarifications and additional experiments adequately address your concerns. Please let us know if any aspect remains unclear. We would be glad to provide further details.

---

### Official Review · Reviewer_oQCt · 2025-10-30

**Soundness:** 2
**Presentation:** 3
**Contribution:** 3
**Rating:** 4
**Confidence:** 3

**Summary:**

This paper revisits Sharpness-Aware Minimization (SAM) and provides a new interpretation of its mechanism: the gradient at the ascent point more accurately approximates the direction toward the worst-case loss in a neighborhood than the local gradient. Motivated by visualization and theoretical analysis showing that this approximation is often inaccurate and deteriorates under multi-step ascent, the authors propose XSAM, which explicitly estimates the ascent direction within a two-dimensional subspace via spherical interpolation. XSAM dynamically adjusts the interpolation factor and updates parameters in the opposite direction to escape local maximum more effectively. Extensive experiments validate the effectiveness of the proposed method.

**Strengths:**

- The paper is generally well-written and easy to follow.
- The authors perform visualization studies to show how single-step SAM gradient directions better approximate ascent directions within the neighborhood, while multi-step SAM may degrade. These visualizations ground the theoretical intuition in empirical phenomena.
- Despite the additional probing steps, the runtime overhead remains negligible. The method is compatible with SAM , making it practical and easy to integrate into real-world training pipelines.

**Weaknesses:**

- The underlying motivation for using $-v(\alpha^*)$ as the final gradient descent direction remains unclear. Following the direction of $-v(\alpha^*)$ appears to encourage moving away from a local neighborhood maximum. However, this does not necessarily guarantee convergence toward a flatter minimum. Additional clarification and theoretical justification would strengthen the argument.
- In the experimental section, the paper primarily compares the proposed method against standard SAM and SGD. Given that numerous SAM variants have been discussed in the related work, including stronger SAM-based baselines would provide a more rigorous and convincing empirical evaluation of the proposed method's effectiveness.

- More representative SAM-based methods, such as GSAM [1], GAM [2], ImbSAM [3], CC-SAM [4], and Focal-SAM [5] should be included in the related work for a more comprehensive review.

-----

[1] Surrogate Gap Minimization Improves Sharpness-Aware Training, ICLR 2022

[2] Gradient Norm Aware Minimization Seeks First-Order Flatness and Improves Generalization, CVPR 2023

[3] ImbSAM: A Closer Look at Sharpness-Aware Minimization in Class-Imbalanced Recognition, ICCV 2023

[4] Class-Conditional Sharpness-Aware Minimization for Deep Long-Tailed Recognition, CVPR 2023

[5] Focal-SAM: Focal Sharpness-Aware Minimization for Long-Tailed Classification, ICML 2025

**Questions:**

Please see above.

---

> ### Author Response · Authors · 2025-11-24
> **Response to Reviewer oQCt**
>
> Thank you for your time and effort in reviewing our work. We sincerely appreciate your recognition of the clarity, visualizations, and applicability of our work. We address the questions and concerns you raised as follows.
>
> **W1: The underlying motivation for using $-v(\alpha^\*)$ as the final gradient descent direction remains unclear. Following the direction of $-v(\alpha^\*)$ appears to encourage moving away from a local neighborhood maximum. However, this does not necessarily guarantee convergence toward a flatter minimum.**
>
> - Moving away from the local neighborhood maximum would directly minimize the maximum loss within the neighborhood, which is the goal and primal objective of SAM.
>
> - Theorem 1 in [1] implies that the generalization error can be bounded by the maximum loss in the local neighborhood.
>
> - The maximum loss in the local neighborhood is, in fact, a more direct metric for generalization. Flatness alone does not guarantee a small error; it only becomes meaningful when the loss is also low. On the other hand, the low loss and the flatness are inherently and jointly captured by the maximum loss in the local neighborhood.
>
> - Conceptually, minimizing the maximum loss in the local neighborhood would lead to a flat minimum. In particular, if the maximum loss approaches the lower bound of the loss, then every point within the neighborhood attains this lower bound. In this case, the loss landscape within the neighborhood becomes perfectly flat.
>
> - In Appendix E.4, we evaluate the flatness of the obtained minimum via the Hessian spectrum (formerly Table 12, now Table 16), visualizations of the loss landscape (formerly Figure 12, now Figure 15), and average sharpness (formerly Figure 12, now Figure 16), respectively. The results consistently demonstrate that updating with $-v(\alpha^\*)$ leads to flatter minima than SGD and SAM.
>
> [1] Foret, Pierre, et al. "Sharpness-aware Minimization for Efficiently Improving Generalization." International Conference on Learning Representations, 2021.
>
> **W2: Given that numerous SAM variants have been discussed in the related work, including stronger SAM-based baselines would provide a more rigorous and convincing empirical evaluation of the proposed method's effectiveness.**
>
> Thank you for your suggestion.
>
> - We have additionally compared XSAM with ASAM, VaSSO, and WSAM on CIFAR-100 using both ResNet-18 and DenseNet-121. As shown in the table below, XSAM achieves the highest accuracy across both architectures, further demonstrating its effectiveness. The results and related details have been incorporated into the revised manuscript in Appendix E.1 (highlighted in blue).
>
> - We note that the techniques in some baselines, e.g., ASAM and VaSSO, are orthogonal to our work, and could potentially be combined with XSAM for further improvements.
>
> - CIFAR-100 on ResNet-18 and DenseNet-121:
> | | | |
> |:---:|:---:|:---:|
> | Method | ResNet-18 | DenseNet-121 |
> | SAM    | 80.93 $\pm$ 0.11 | 83.81 $\pm$ 0.02 |
> | ASAM   | 81.11 $\pm$ 0.06 | 83.99 $\pm$ 0.25 |
> | VaSSO  | 80.84 $\pm$ 0.15 | 83.78 $\pm$ 0.25 |
> | WSAM   | 80.95 $\pm$ 0.19 | 83.91 $\pm$ 0.15 |
> | XSAM   | 81.24 $\pm$ 0.07 | 83.96 $\pm$ 0.10 |
> | XSAM + ASAM | **81.68 $\pm$ 0.11** | **84.06 $\pm$ 0.21** |
>
>     #### All baseline methods are carefully tuned for optimal performance. XSAM uses the same $\rho$ as SAM, as in the paper.
>
> **W3: More representative SAM-based methods, such as GSAM, GAM, ImbSAM, CC-SAM, and Focal-SAM, should be included in the related work for a more comprehensive review.**
>
> - Thank you for highlighting these related works. We have additionally discussed them in Section 4 and Appendix G of the revised manuscript (highlighted in blue). Specifically, GSAM and GAM, which define new update rules for sharpness-aware minimization, have been discussed in Section 4 (Related Work). ImbSAM, CC-SAM, and Focal-SAM, which focus on long-tail learning, are outlined in Appendix G (Additional Related Work).
>
> ---
> We sincerely hope that the above clarifications and additional experiments adequately address your concerns. Please let us know if any aspect remains unclear. We would be glad to provide further details.

---

### Official Review · Reviewer_Rjdp · 2025-11-01

**Soundness:** 3
**Presentation:** 2
**Contribution:** 3
**Rating:** 4
**Confidence:** 4

**Summary:**

In this paper, the authors study how the sharpness-aware minimization (SAM) algorithm works and how to improve it.
They first show that the single-step ascent used in SAM, when applied to current parameters, is a better approximation of the maximum over a ball, compared to the naive gradient. This shows why the SAM algorithm is superior in practice, as it is indeed minimizing the maximum loss over a ball.
Indeed, they show that the approximation of max loss with naive gradient in a single step is often inaccurate, and the quality gets worse if we consider multi-step ascent, a phenomenon observed in practice.


In Proposition 1, they show that while SAM provides a better approximation of the maximum loss around parameters (up to a ball), linear combinations of SAM updates and naive gradients can be even better. They thus propose XSAM:
They probe in the hyperplane of the naive gradient and the SAM gradient, using spherical linear interpolation (Eq. 6). It needs some maximization over a parameter alpha.
The proposed method proposed only a small additional computational overhead.
They conclude the paper with experiments supporting their proposed method.

**Strengths:**

- The theoretical work on explaining the success of the SAM algorithm in minimizing the SAM objective, compared to the naive gradient, is of potential community interest.


 - The paper is probing between the SAM and naive gradient method, and it shows that it will strictly improve the performance, which is of potential interest to the community. The experiments also support this.

**Weaknesses:**

- Long sentences, hard to follow. The paper would benefit from better writing, focusing on short and clean sentences.

**Questions:**

This is an interesting paper, and I believe it makes a good contribution to our understanding of why SAM works and how to improve it.

My only concern is that the paper lacks good writing. I appreciate that the authors provided an explanation of how they used LLMs just for polishing sentences, but the fact that there are long sentences in the paper is quite annoying (sometimes LLMs suggest them as a way to polish the paper, but they indeed make reading the paper harder).  For example, Line 19-24 of abstract, the main question of the paper, is introduced in a very, very long sentence, hard to follow. Please modify the abstract and other parts of the paper, and make sure the message is delivered clearly. The paper, in my opinion, needs major changes.


- (minor) Line 11 in the algorithm: Where is the definition of v(alpha) within the algorithm?

---

> ### Author Response · Authors · 2025-11-24
> **Response to Reviewer Rjdp**
>
> Thank you for your time and effort in reviewing our work. We sincerely appreciate your recognition of our contributions toward understanding and improving SAM, and are encouraged to hear that you find our work interesting. We address the questions and concerns you raised as follows.
>
> ---
> **W1 & Q1: My only concern is that the paper lacks good writing. Please modify the abstract and other parts of the paper, and make sure the message is delivered clearly.**
>
> - Thank you for the constructive feedback. We agree that some sentences were overly long. To address this, we have thoroughly revised the manuscript to improve clarity and readability. In particular, long and complex sentences have been simplified. Key revised sentences are highlighted in magenta in the revised manuscript.
>
> - For your convenience, examples of the revisions (from the abstract) are provided below.
>     - **Sentence 1:**
>         - **Original (One sentence):** Although this practice is justified as approximately optimizing the objective by neglecting the (full) derivative of the ascent point with respect to the current parameters, a direct and intuitive understanding of why using the gradient at the ascent point to update the current parameters works superiorly (despite a shift in location) is still lacking.
>         - **Revised (Two sentences):** This practice can be justified as approximately optimizing the objective by neglecting the (full) derivative of the ascent point with respect to the current parameters. Nevertheless, a direct and intuitive understanding of why using the gradient at the ascent point to update the current parameters works superiorly, despite being computed at a shifted location, is still lacking.
>     - **Sentence 2:**
>         - **Original (One sentence):** Our work bridges this gap by proposing and justifying a novel, intuitive interpretation: the gradient at the single-step ascent point, when applied to the current parameters, provides a better approximation of the direction from the current parameters towards the maximum within the local neighborhood than the local gradient, thereby enabling a more direct escape from the maximum within the local neighborhood.
>         - **Revised (Three sentences):** Our work bridges this gap by proposing a novel and intuitive interpretation. We show that the gradient at the single-step ascent point, when applied to the current parameters, provides a better approximation of the direction from the current parameters toward the maximum within the local neighborhood than the local gradient. This improved approximation thereby enables a more direct escape from the maximum within the local neighborhood.
>
> ---
> **Q2: Line 11 in the algorithm. Where is the definition of $v(\alpha)$ within the algorithm?**
>
> - Thank you for catching this issue. The definition of $v(\alpha)$ has now been added to line 12 of Algorithm 1.
>
> ---
> We sincerely hope that our revisions adequately address your concerns regarding the writing. Please let us know if you find any aspect that requires additional refinement. We would be glad to further improve the clarity and readability of the manuscript.

---

### Official Review · Reviewer_Qt3b · 2025-11-04

**Soundness:** 4
**Presentation:** 3
**Contribution:** 3
**Rating:** 6
**Confidence:** 3

**Summary:**

This paper revisits Sharpness-Aware Minimization (SAM) and introduces eXplicit Sharpness-Aware Minimization (XSAM), which improves upon SAM by explicitly estimating the direction toward the local maximum loss within a neighborhood, rather than relying on the potentially inaccurate gradient at the ascent point. XSAM consistently outperforms SAM and its variants across various models and datasets in both single-step and multi-step settings. The authors provide both theoretical justification and empirical visualizations to explain why SAM’s approximation can be suboptimal and how XSAM addresses these limitations.

**Strengths:**

1. XSAM provides a more accurate and adaptive estimation of the direction toward the local maximum, leading to better generalization.

2. The method is unified across single-step and multi-step settings and shows consistent improvements over SAM with minimal computational overhead.

3. The paper offers a clear theoretical and intuitive explanation of SAM’s limitations, enhancing understanding of sharpness-aware optimization.

**Weaknesses:**

1. XSAM introduces additional hyperparameters (e.g., search range for α and update frequency), which may complicate hyperparameter tuning.

2. Although the overhead is small, XSAM still requires extra forward passes for direction estimation, slightly increasing computational cost.

3. Although the paper critiques multi-step SAM’s degradation, the comparison with existing multi-step variants (like MSAM or LSAM) could be more extensive to fully establish XSAM’s superiority in that regime.

**Questions:**

1. Your 2-D search is restricted to the plane spanned by v_{0} and v_{1}.
What theoretical or empirical evidence guarantees that the true worst-case perturbation   \delta^{*} (i.e., the arg-max of L(\theta+\delta) inside the \rho-ball) actually lies in this plane, and how might the method behave if the Hessian is highly anisotropic or the loss landscape is strongly non-quadratic?

---

> ### Author Response · Authors · 2025-11-24
> **Response to Reviewer Qt3b, Part 1/2**
>
> Thank you for your time and effort in reviewing our work. We sincerely appreciate your recognition of the soundness of our work and its enhanced understanding of sharpness-aware optimization. We address the questions and concerns you raised as follows.
>
> **Q1-1: What theoretical or empirical evidence guarantees that the true worst-case perturbation actually lies in the 2D search plane spanned by v_0 and v_1?**
>
> - In fact, it is not guaranteed that the worst-case perturbation lies in the 2D search plane. And it is virtually impossible to guarantee that the worst-case perturbation lies in any low-dimensional subspace, given the highly nonlinear property of neural networks.
>
> - It is guaranteed, as by the second statement of Proposition 1, that a point with a higher loss than that found by the SAM gradient lies within the 2D search plane. It is, in fact, further guaranteed by construction that the ray defined by $\vartheta_k$ and $g_k$ lies in the 2D search plane, which ensures that the point with the highest known loss (identified by the $k$ ascent steps) lies in the 2D search plane. This is almost the best we can do in a 2D search plane, given only the information provided by these $k$ ascent steps.
>
> - A higher-dimensional search space could, in principle, provide stronger guarantees, but the computational cost would grow exponentially. Since identifying the exact worst-case perturbation is practically infeasible, we believe that restricting the search to a 2D plane offers a favorable balance between computational efficiency and effectiveness.
>
> **Q1-2: How might the method behave if the Hessian is highly anisotropic or the loss landscape is strongly non-quadratic?**
>
> - Our guarantee that (i) a point with a higher loss than that found by the SAM gradient lies in the 2D search plane, and (ii) the point with the highest known loss (identified by the ray of $\vartheta_k$ and $g_k$) lies in the 2D search plane, DOES NOT rely on the quadratic assumption of the loss landscape.
>
> - The quadratic assumption in Proposition 1 is only one of the feasible ways to ensure that the point identified by the SAM gradient is not the worst-case point. We use such a theoretical result to affirm that the approximation by the SAM gradient is indeed typically suboptimal, even in the quadratic case.
>
> - Our extensive visualizations have clearly demonstrated that a better approximation than SAM generally exists in the 2D search plane. So, XSAM can generally better identify and escape the nearby high-loss region and thereby lead to better generalization.

---

> > ### Author Response · Authors · 2025-11-24
> > **Response to Reviewer Qt3b, Part 2/2**
> >
> > **W1: XSAM introduces additional hyperparameters (e.g., search range for $\alpha$ and update frequency), which may complicate hyperparameter tuning.**
> >
> > While XSAM introduces a few additional hyperparameters, they are quite robust and insensitive in practice.
> >
> > - In our experiments, we never observe $\alpha^\*$ exceeding the range $[0.0, 4.0]$. We use a default search range of $[0.0, 2.0]$ in our experiments, which is typically sufficient. Nevertheless, for a more conservative choice, one may extend it to $[0.0, 4.0]$, which merely doubles the negligible overhead of searching $\alpha^\*$ -- which remains negligible.
> >
> > - As shown in Figure 4, XSAM is insensitive to the update frequency of $\alpha^\*$. In fact, we have never tuned this parameter in our typical experiments and, by default, set it to update once per epoch.
> >
> > **W2: Although the overhead is small, XSAM still requires extra forward passes for direction estimation, slightly increasing computational cost.**
> >
> > - As shown in Table 1 and Appendix C, the additional computational cost is indeed negligible.
> >
> > - Importantly, this negligible computational cost yields substantial accuracy gains.
> >
> > **W3: Although the paper critiques multi-step SAM’s degradation, the comparison with existing multi-step variants could be more extensive to fully establish XSAM’s superiority in that regime.**
> >
> > - We have more experiments in the appendix, where Table 11 (now Table 12 in the revised manuscript) compares XSAM and multi-step SAM variants under varying $\rho$.
> >
> > - In response to your feedback, we have further compared XSAM with MSAM and LSAM under $k=1, 2, 4$ on DenseNet-121 using CIFAR-100 and on ResNet-18 using CIFAR-10.
> >
> > - We summarize the multi-step experiment results in the tables below. As we can see, XSAM consistently attains high accuracy while maintaining strong robustness. In contrast, existing multi-step SAM variants may even underperform their single-step counterparts. The additional results have been incorporated into Appendix E.2 of the revised manuscript (highlighted in blue).
> >
> > - ResNet-18, CIFAR-100, varying $k$:
> > | | | | |
> > |:------:|:-----------------:|:-----------------:|:-----------------:|
> > | Method | k = 1             | k = 2             | k = 4             |
> > | SAM    | 80.93 $\pm$ 0.11  | 80.91 $\pm$ 0.10  | 80.65 $\pm$ 0.26  |
> > | LSAM   | 80.93 $\pm$ 0.11  | 80.94 $\pm$ 0.09  | 80.74 $\pm$ 0.18  |
> > | LSAM+  | 80.61 $\pm$ 0.20  | 80.83 $\pm$ 0.11  | 80.41 $\pm$ 0.03  |
> > | MSAM   | 80.93 $\pm$ 0.11  | 81.18 $\pm$ 0.06  | 81.01 $\pm$ 0.09  |
> > | MSAM+  | 80.83 $\pm$ 0.05  | 80.86 $\pm$ 0.34  | 80.77 $\pm$ 0.08  |
> > | XSAM   | **81.27 $\pm$ 0.07**  | **81.44 $\pm$ 0.09**  | **81.37 $\pm$ 0.24**  |
> >
> > - ResNet-18, CIFAR-100, varying $\rho$, $k = 3$:
> > | | | | |
> > |:------:|:------------------:|:------------------:|:------------------:|
> > | Method | ρ = 0.04     | ρ = 0.08     | ρ = 0.12      |
> > | SAM    | 80.79 $\pm$ 0.41   | 80.75 $\pm$ 0.27   | 79.72 $\pm$ 0.33   |
> > | LSAM   | 81.00 $\pm$ 0.21   | 81.20 $\pm$ 0.24   | 81.16 $\pm$ 0.04   |
> > | LSAM+  | 80.56 $\pm$ 0.20   | 80.77 $\pm$ 0.04   | 80.21 $\pm$ 0.27   |
> > | MSAM   | 81.04 $\pm$ 0.06   | 81.12 $\pm$ 0.17   | 80.93 $\pm$ 0.11   |
> > | MSAM+  | 80.72 $\pm$ 0.16   | 81.16 $\pm$ 0.05   | 81.16 $\pm$ 0.05   |
> > | XSAM   | **81.23 $\pm$ 0.06**   | **81.36 $\pm$ 0.08**   | **81.29 $\pm$ 0.06**   |
> >
> > - DenseNet-121, CIFAR-100, varying $k$:
> > | | | | |
> > |:------:|:-----------------:|:----------------:|:----------------:|
> > | Method | k = 1             | k = 2            |k = 4             |
> > | LSAM   | 83.81 $\pm$ 0.02  | 83.82 $\pm$ 0.28 |83.40 $\pm$ 0.17  |
> > | MSAM   | 83.81 $\pm$ 0.02  | 83.67 $\pm$ 0.23 |83.74 $\pm$ 0.18  |
> > | XSAM   | **83.96 $\pm$ 0.10**  | **84.02 $\pm$ 0.31** | **84.05 $\pm$ 0.04** |
> >
> > - ResNet-18, CIFAR-10, varying $k$:
> > | | | | |
> > |:------:|:----------------:|:----------------:|:----------------:|
> > | Method | k = 1            | k = 2            | k = 4            |
> > | LSAM   | 96.59 $\pm$ 0.06 | 96.66 $\pm$ 0.03 | 96.72 $\pm$ 0.07 |
> > | MSAM   | 96.59 $\pm$ 0.06 | 96.78 $\pm$ 0.05 | 96.80 $\pm$ 0.07 |
> > | XSAM   | **96.74 $\pm$ 0.04** | **96.81 $\pm$ 0.06** | **96.81 $\pm$ 0.11** |
> >
> > ---
> > We sincerely hope that the above clarifications and additional experiments adequately address your concerns regarding our method. Please let us know if any aspect remains unclear. We would be glad to provide further details.

---

### Author Response · Authors · 2025-12-02
**Summary of Reviews and Responses**

Dear Chairs,

Thank you for taking the time to evaluate our work.

Given the special circumstance, we sincerely appreciate the Area Chair’s additional effort. For your convenience, we summarize the reviews and our responses as follows (using review codes such as W1/Q1 for cross-reference):
\
&nbsp;
- Overall / Ratings:
    - Our paper received initial ratings of **6, 4, 4, 4** from reviewers Qt3b, Rjdp, oQCt, and SZGB, respectively.
    - *Rjdp* (rating 4) states that **"my only concern is that the paper lacks good writing"**: some sentences are overly long.
    \
    &nbsp;
- Recognitions:
    - *Qt3b* considers the **soundness** of our paper as **excellent** and states that **"the paper offers a clear theoretical and intuitive explanation of SAM’s limitations, enhancing understanding of sharpness-aware optimization"**.
    \
    &nbsp;
    - *Rjdp* states that **"this is an interesting paper, and I believe it makes a good contribution to our understanding of why SAM works and how to improve it"**.
    \
    &nbsp;
    - *oQCt* states that **"the paper is generally well-written and easy to follow"** and **"these visualizations ground the theoretical intuition in empirical phenomena"**.
    \
    &nbsp;
    - *SZGB* states that **"this paper is well-motivated and easy to follow"**, **"the authors identified a critical issue in estimating the gradient descent direction"**, and **"the authors have presented their work in a good shape"**.
    \
    &nbsp;
- Concerns and Responses:
    - *Qt3b* is concerned that
        - **Introduces additional hyperparameters.** (W1)
            - Clarified: typically fixed and requiring no tuning.
        - **Slightly increasing computational cost.** (W2)
            - Clarified: the additional cost is negligible, whereas the gain is substantial.
        - **Comparison with existing multi-step variants could be more extensive.** (W3)
            - Addressed via additional experiments: XSAM consistently attains the highest accuracy.
        - **Whether it is guaranteed that the true worst-case perturbation lies in the 2D search plane.** (Q1)
            - Clarified: it is guaranteed that (i) a point with a higher loss than that found by SAM lies in, and (ii) the point with maximum known loss lies in.
        \
        &nbsp;
    - *Rjdp* is concerned that
        - **Some sentences are overly long.** (W1 & Q1)
            - Addressed via breaking long sentences down.
        - **The definition of v(alpha) is missing within the Algorithm.** (Q2)
            - (Minor issue) Added its definition to Algorithm 1.
        \
        &nbsp;
    - *oQCt* is concerned that
        - **Moving away from a local neighborhood maximum does not necessarily guarantee convergence toward a flatter minimum.** (W1)
            - Clarified: moving away from the maximum will minimize the maximum loss within the local neighborhood, which is directly related to better generalization and flatness.
        - **Including stronger SAM-based baselines would provide a more rigorous and convincing empirical evaluation.** (W2)
            - Addressed via additional experiments: XSAM still achieves the highest accuracy.
        - **More representative SAM-based methods should be included in the related work.** (W3)
            - Included.
        \
        &nbsp;
    - *SZGB* is concerned that
        - **Why XSAM enables a more direct escape from the maximum within the local neighborhood.** (W1)
            - Clarified: since it better approximates the direction from the current parameter toward the maximum.
        - **The loss peak in the 2D hyperplane might vary significantly as $\rho$ changes.** (W2)
            - Clarified: we implicitly assume that the high-loss region remains stable across different 2D slices; claims are supported beyond the visual evidence.
        - **Why, for sufficiently large $\rho_m$, the $\rho_m^2$ term dominates.** (W3)
            - Clarified: it is assumed to be essentially quadratic.
        - **More baselines and datasets should be included.** (W4-1, W4-3)
            - Addressed via additional experiments: XSAM consistently achieves the highest performance.
        - **How it performs when rescaled with the base gradient $\|g_0\|$.** (W4-2)
            - Addressed via content in Appendix F: the effectiveness varies across models and settings.
        \
        &nbsp;
- Revisions:
    - Revised sentences (that were previously long) are highlighted in magenta.
    - Other new content is highlighted in blue.
    \
    &nbsp;

We hope that our clarifications and additional experiments have sufficiently addressed the concerns. We sincerely appreciate your consideration.

Sincerely,

Authors

---

### Meta-Review · Area_Chair_XHA4 · 2026-01-09

**Summary:**

The paper proposes Explicit Sharpness-Aware Minimization (XSAM) to address the limitations of the gradient approximation used in standard SAM and its multi-step variants. The authors provide a novel interpretation that the single-step ascent gradient provides a better, though still inaccurate, approximation of the direction toward the maximum loss than the local gradient. XSAM remedies this by explicitly searching for the ascent direction within a dynamic 2D subspace. Initial reviewer concerns focused heavily on the writing style (overly long sentences), the sufficiency of baselines (requests for ASAM, MSAM, and recent variants), theoretical justifications for the search space and update rules, and the lack of robustness evaluations. The authors provided a comprehensive rebuttal, including a revised manuscript to improve readability, extensive new experimental comparisons with suggested baselines, and additional analyses of loss landscapes and robustness. Given the depth of the rebuttal and the widespread empirical improvements shown, the paper is in a strong position and I recommend accept.

**Reviewer Concerns:**

*** ADDRESSED

Writing Quality (Reviewer Rjdp): The reviewer’s primary concern was the readability of the paper, specifically identifying "overly long" sentences. The authors addressed this by thoroughly revising the manuscript (highlighting changes) and breaking down complex sentences, specifically in the abstract and introduction.

Baselines and Comparisons (Reviewers Qt3b, oQCt, SZGB): Multiple reviewers requested comparisons against stronger and more relevant baselines. The authors responded by adding experiments comparing XSAM against ASAM, VaSSO, WSAM, MSAM, and LSAM on CIFAR-100 (ResNet-18 and DenseNet-121). The results demonstrated XSAM's consistent superiority.

Robustness (Reviewer SZGB): The reviewer asked for performance on corrupted datasets. The authors added experiments on CIFAR-10-C and CIFAR-100-C, showing XSAM maintains robustness.

Mechanism Clarification (Reviewer oQCt): The reviewer questioned if moving away from a local maximum guarantees convergence to a flatter minimum. The authors provided additional analyses, including Hessian spectrum calculations and loss landscape visualizations, to empirically demonstrate that XSAM leads to flatter minima.

*** OUTSTANDING

Theoretical Rigor (Reviewer SZGB): While the authors provided empirical plots to support Proposition 1, the reviewer's concern regarding the "hand-wavy" nature of the proof (specifically the reliance on sufficiently large $\rho_m$ and the quadratic assumption) remains a theoretical limitation. The authors acknowledge these are idealized assumptions, though they argue they are sufficient for establishing the motivation.

Search Space Guarantee (Reviewer Qt3b): The concern regarding whether the true worst-case perturbation lies within the defined 2D search plane is theoretically valid. The authors clarified that while not guaranteed, the plane contains the best known information, but this remains a heuristic constraint rather than a proven optimal subspace.

**Reviewer Scores:**

Reviewer Qt3b (Current: 6): This reviewer was already positive. Since their main weakness regarding the extent of multi-step comparisons was addressed with new data showing XSAM outperforms MSAM and LSAM, they would likely maintain or slightly increase their score to a strong accept.

Reviewer Rjdp (Current: 4): This reviewer’s score was explicitly tied to "lack of good writing." Since the authors performed a major revision specifically targeting the sentence structure issues raised, this reviewer would likely raise their score to a 5 or 6, as their "only concern" was addressed.

Reviewer oQCt (Current: 4): This reviewer requested stronger baselines (ASAM, etc.) and clarification on the motivation. The authors provided exactly the requested baseline comparisons and additional flatness analysis. With these comprehensive additions, it is likely this reviewer would move to a positive score (5 or 6).

Reviewer SZGB (Current: 4): This reviewer had the most technical list of demands, including new baselines, scaling inquiries, and corruption benchmarks. The authors executed these additional experiments (CIFAR-C, ASAM comparisons) and discussed gradient scaling in the appendix. While the theoretical proof concerns might prevent a very high score, the empirical rigor of the rebuttal should likely move this reviewer to a 5 or 6.

---

### Decision · Program_Chairs · 2026-01-26

Accept (Poster)